# Dinucleoside polyphosphates act as 5′-RNA caps in bacteria

Oldřich Hudeček[1,3], Roberto Benoni[1,3], Paul E. Reyes-Gutierrez[1], Martin Culka [1], Hana Šanderová[2], Martin Hubálek[1], Lubomír Rulíšek [1], Josef Cvačka [1], Libor Krásný [2] & Hana Cahová [1✉]

It has been more than 50 years since the discovery of dinucleoside polyphosphates ($Np_nNs$) and yet their roles and mechanisms of action remain unclear. Here, we show that both methylated and non-methylated $Np_nNs$ serve as RNA caps in *Escherichia coli*. $Np_nNs$ are excellent substrates for T7 and *E. coli* RNA polymerases (RNAPs) and efficiently initiate transcription. We demonstrate, that the *E. coli* enzymes RNA 5′-pyrophosphohydrolase (RppH) and bis(5′-nucleosyl)-tetraphosphatase (ApaH) are able to remove the $Np_nN$-caps from RNA. ApaH is able to cleave all $Np_nN$-caps, while RppH is unable to cleave the methylated forms suggesting that the methylation adds an additional layer to RNA stability regulation. Our work introduces a different perspective on the chemical structure of RNA in prokaryotes and on the role of RNA caps. We bring evidence that small molecules, such as $Np_nNs$ are incorporated into RNA and may thus influence the cellular metabolism and RNA turnover.

[1] Institute of Organic Chemistry and Biochemistry of the Czech Academy of Sciences, Flemingovo nam. 2, 16610, Prague 6, Czech Republic. [2] Institute of Microbiology of the Czech Academy of Sciences, Vídeňská 1083, 142 20, Prague, Czech Republic. [3]These authors contributed equally: Oldřich Hudeček, Roberto Benoni. ✉email: cahova@uochb.cas.cz

The role and chemical structure of the 5′-end of prokaryotic RNA is still unclear. The discovery of nicotinamide adenine dinucleotide (NAD)[1,2] and coenzyme A (CoA)[3] 5′-RNA caps changed the perception of the RNA structure. 5′-caps are usually cleaved by NudiX enzymes[4] (NudC (refs. [1,5,6]), RNA 5′-pyrophosphohydrolase (RppH)[7,8]), which can, besides their decapping role (eukaryotic Nudt16 and Dcp2 (refs. [9,10])), also cleave nucleoside diphosphates linked to another moiety (e.g., dinucleoside polyphosphates (Np$_n$Ns)[11,12]). Np$_n$Ns are ubiquitous molecules[13–15] present in both prokaryotic and eukaryotic organisms. Their intracellular concentrations can increase from the μM to the mM range under stress conditions[11,16], which is why they are often called alarmones. Nevertheless, the molecular targets of the alarm signaled by Np$_n$Ns have not been identified yet. The main source of the Np$_n$Ns is the back reaction of an aminoacyl adenylate with an acceptor nucleotide catalyzed by various aminoacyl-tRNA synthetases[17]. But there is evidence that e.g., the ubiquitinylation process can lead to Ap$_4$A or Ap$_3$A (ref. [18]). Even though they have a similar chemical structure to known RNA caps, such as NAD (refs. [1,19,20]) and the 7-methylguanylate cap[21], they were never detected as a part of RNA.

In this work, we show that Np$_n$Ns can be accepted by two types of RNA polymerases (RNAPs; bacteriophage T7 and E. coli) as non-canonical initiating nucleotides (NCINs) in in vitro transcription. To prove the existence of Np$_n$N-RNA caps in vivo, we develop an liquid chromatography–mass spectrometry (LC–MS) technique for the detection of Np$_n$Ns in isolated and digested RNA. We detect six previously unknown Np$_n$N-caps (Ap$_3$A, m$^6$Ap$_3$A, Ap$_3$G, m$^7$Gp$_4$Gm, Ap$_5$A, and mAp$_5$G) in fractions of short RNA (sRNA) from Escherichia coli harvested in exponential phase and additional three Np$_n$N-caps (mAp$_4$G, mAp$_5$A, and 2mAp$_5$G) in late stationary phase. Some of the detected Np$_n$N-caps are mono- or dimethylated. We identify two enzymes, RppH from the NudiX family and bis(5′-nucleosyl)-tetraphosphatase (ApaH), as decapping enzymes that can cleave Np$_n$N-RNA. While the methylations of Np$_n$N-caps protect the RNA from decapping by RppH, ApaH cleaves even the methylated forms of Np$_n$N-caps. We propose that bacteria use methylated caps as protection against RNA degradation under starvation conditions.

## Results

**In vitro incorporation of Np$_n$Ns into RNA.** To investigate whether Np$_n$Ns (Fig. 1a) can serve as NCINs similarly to NAD and CoA (ref. [22]), we performed in vitro transcriptions in the presence of different Np$_n$Ns (Ap$_{3-6}$A, Ap$_{4-5}$G, and Gp$_4$G, Fig. 1b) with T7 RNAP (single peptide chain enzyme, derived from an E. coli-infecting phage) and E. coli RNAP (multi-subunit enzyme). T7 RNAP was selected as a first tool to explore the capability of Np$_n$Ns to be substrates of RNAPs as it was previously shown to be able to use comparably sized non-canonical initiating substrates[23]. Consistently, based on three-dimensional structures, the nucleotide-binding pockets for initiation phase of T7 (ref. [24]) and E. coli[22] RNAP are spacious enough to accommodate such substrates.

The in vitro transcription produced a mixture of capped and uncapped RNA (step 1 in Fig. 1b, Supplementary Fig. 1a). The presence of the caps was confirmed by electrophoretic analysis after treatments with the 5′-polyphosphatase and the Terminator™ 5′-phosphate-dependent exonuclease (terminator, steps 2 and 3 in Fig. 1b). The former enzyme dephosphorylated the 5′-triphosphate RNA (5′-ppp RNA) but not the capped RNA. The terminator then digested all RNA with 5′-monophosphate termini (5′-p RNA) and left the capped RNA intact. We observed that all tested Np$_n$Ns were excellent substrates for the T7 RNAP

and served as NCINs for the in vitro transcription (Fig. 1c). The only enzymatic incorporation of an Np$_n$N into RNA that had been previously reported involved methylated derivates of Gp$_n$G (diguanosines polyphosphates, eukaryotic cap variants) prepared for a translation inhibition study[25].

To identify the best substrate for T7 RNAP under in vitro conditions, we varied the concentrations of the Np$_n$Ns in the presence of constant (1 mM) ATP and GTP concentrations (Fig. 1e, Supplementary Fig. 1c, d). The amount of capped RNA increased linearly with the concentration of Np$_n$Ns. When the ratio of ATP (GTP) to Np$_n$N was 1, we observed between 27% (for Ap$_6$A) and 46% (for Ap$_4$G) of capped products. The NAD-capped RNA was produced in a comparable amount to the majority of Np$_n$Ns. The similar behavior of Np$_n$Ns to NAD supports the theory that these molecules might be present as 5′-RNA caps in cells.

Subsequently, we tested the E. coli RNAP that is known to accept NAD as an NCIN both in vitro and in vivo[22,26,27] (Supplementary Fig. 1b). To confirm the existence of 5′-capped RNA products, we treated them with the 5′-polyphosphatase and the terminator. We used two different models of well characterized promoters (rrnB P1 and rnaI) for E. coli RNAP. In both cases, we observed a higher production of Np$_n$N-RNA compared to NAD-RNA (Fig. 1d). Nevertheless, the promoter sequence may affect the efficiency of incorporation of Np$_n$Ns, the same way it influences the incorporation of NAD (ref. [22]).

**LC–MS analysis of E. coli sRNA.** Next, we wanted to determine whether Np$_n$Ns exist as 5′-RNA caps in vivo in E. coli. We established an LC–MS method for their detection in RNA. As the intracellular concentration of Np$_n$Ns is known to grow under stress conditions[11,16,28], we collected cells from exponential (EXP, OD = 0.3) and late stationary (STA) phases of growth. We focused on sRNA where the NAD-cap[2] and the CoA-cap[3] have also been detected. The purified sRNA was washed to remove all non-covalently interacting molecules and digested by Nuclease P1 into the form of nucleotides (Fig. 2a). The negative control samples, where the addition of Nuclease P1 was omitted, did not show any signals of nucleotides or Np$_n$Ns, which excluded the possibility of non-covalently bound contamination (Supplementary Figs. 2–4).

In all the digested sRNA, we observed signals of Ap$_3$A ([M−H]$^-$ at $m/z$ 755.077), Ap$_3$G ([M−H]$^-$ at $m/z$ 771.071), and Ap$_5$A ([M−2H]$^{2-}$ at $m/z$ 457.009; Supplementary Fig. 2a–c). We also observed strong signals of mono- and dimethylated forms of Np$_n$Ns, specifically methyl-Ap$_3$A ([M−H]$^-$ at $m/z$ 769.077), dimethyl-Gp$_4$G ([M−2H]$^{2-}$ at $m/z$ 447.017), and methyl-Ap$_5$G ([M−2H]$^{2-}$ at $m/z$ 472.019; Supplementary Fig. 3a–c). Besides the previously mentioned caps, we detected signals of methyl-Ap$_4$G ([M−2H]$^{2-}$ at $m/z$ 432.019), methyl-Ap$_5$A ([M−2H]$^{2-}$ at $m/z$ 463.992), and dimethyl-Ap$_5$G ([M−2H]$^{2-}$ at $m/z$ 479.014; Fig. 2b, Supplementary Fig. 4a–c) in STA. We compared our detected $m/z$ signals with those reported by Liu[2]. The only similar $m/z$ was 771.073, which can correspond to the [M−H]$^-$ of Ap$_3$G.

**LC–MS structure confirmation.** To validate the structure of the detected Np$_n$N-caps, we grew E. coli in minimal media with the sole source of nitrogen from either $^{14}NH_4Cl$ or $^{15}NH_4Cl$. We detected only three Np$_n$N-caps: methyl-Ap$_3$A, Ap$_3$G, and dimethyl-Gp$_4$G (Fig. 2c–f, Supplementary Fig. 5a–c), because this type of growth represents a different type of stress. This experiment confirmed the presence of ten nitrogen atoms in every detected molecule. To further verify the chemical structure of Np$_n$Ns, we compared the LC–MS properties of standard Gp$_4$G with the isomeric p$_3$GpG. While half of the p$_3$GpG was

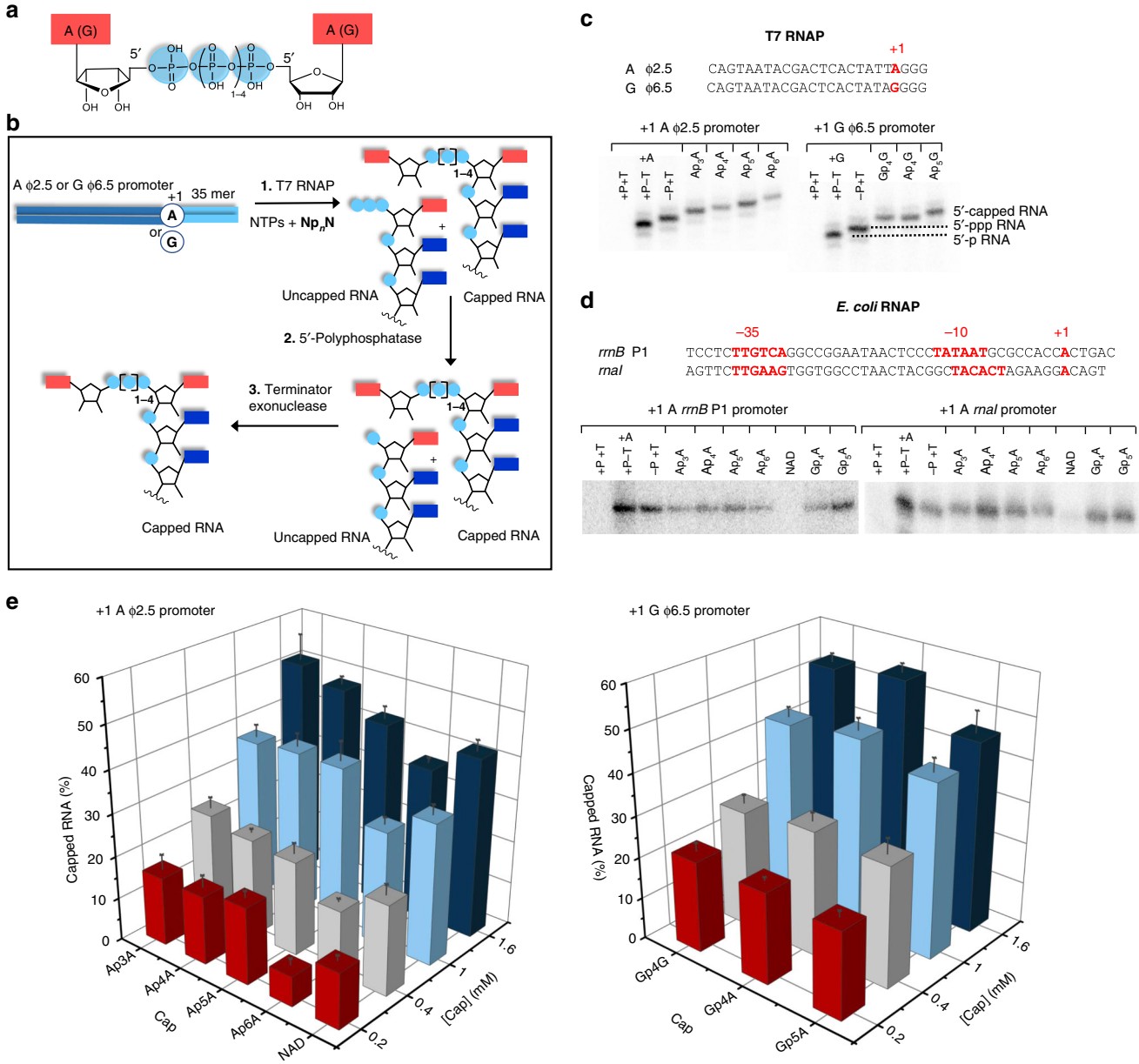

**Fig. 1 Np$_n$Ns are excellent substrates for RNAP. a** The chemical structure of Np$_n$Ns. **b** Scheme of the in vitro transcription with RNAP in the presence of Np$_n$Ns and template DNA yielding RNA starting with A or G. The first step resulted in a mixture of capped and uncapped RNAs, which was then treated by 5'-polyphosphatase (P). In the third step, the 5'-p RNA was degraded by terminator exonuclease (T). **c** Polyacrylamide gel electrophoretic (PAGE—12%) analysis of the (α-$^{32}$P GTP labeled) in vitro transcription products (35 nt) with T7 RNAP followed by P and/or T treatment (all experiments were performed in triplicates). If not specified, samples were treated with both P and T. **d** PAGE analysis of the in vitro transcription products with *E. coli* RNAP, and two templates with promoter *rrnB* P1 and *rnaI*, leading to A starting 144 nt long RNA (*rrnB* P1) or 71 nt long RNA (*rnaI*), followed by P and/or T treatment (all experiments were performed in triplicates). **e** Percentage of different types of capped RNA produced by in vitro transcription with T7 RNAP calculated from PAGE analysis. The depth axis represents various concentrations of Np$_n$Ns (0.2—red, 0.4—gray, 1—light blue, and 1.6 mM—dark blue) at a constant concentration of ATP (1 mM) and GTP (1 mM). The left panel shows the percentage of Ap$_{3-6}$A and NAD-capped RNA, the right panel shows the percentage of Ap$_{4-5}$G and Gp$_4$G. All experiments were performed in triplicates and calculated as average values. Error bars indicate standard deviations. Source data are provided in the Source Data file.

fragmented in the ionization source to p$_2$GpG, the Gp$_4$G stayed intact (Supplementary Fig. 6a, b). The same behavior was observed for dimethyl-Gp$_4$G in the *E. coli* RNA sample, proving the existence of the internal polyphosphate chain. By linear ion trap LC–MS, we detected an intact triphosphate chain of Ap$_3$A confirming its structure (Supplementary Fig. 7).

To identify the methylation positions, the synthetic standards of methylated Np$_n$Ns are necessary to define their retention time and the best fragmentation conditions. As there are many

possible methylation positions in the majority of detected methylated Np$_n$N-caps making their synthesis extremely demanding, we focused on the characterization of 2mGp$_4$G and mAp$_3$A. The LC–MS analysis of *E. coli* sRNA confirmed the presence of one methyl group per one guanosine moiety in 2mGp$_4$G (Supplementary Fig. 8). We hypothesized that these methylations could be in the positions $N^7$ (m$^7$G) and 2'-O (Gm) similarly to eukaryotic m$^7$Gp$_3$Nm RNA caps. We used custom-synthesized m$^7$Gp$_4$Gm, m$^1$Gp$_4$Gm, and m$^2$Gp$_4$Gm. In parallel,

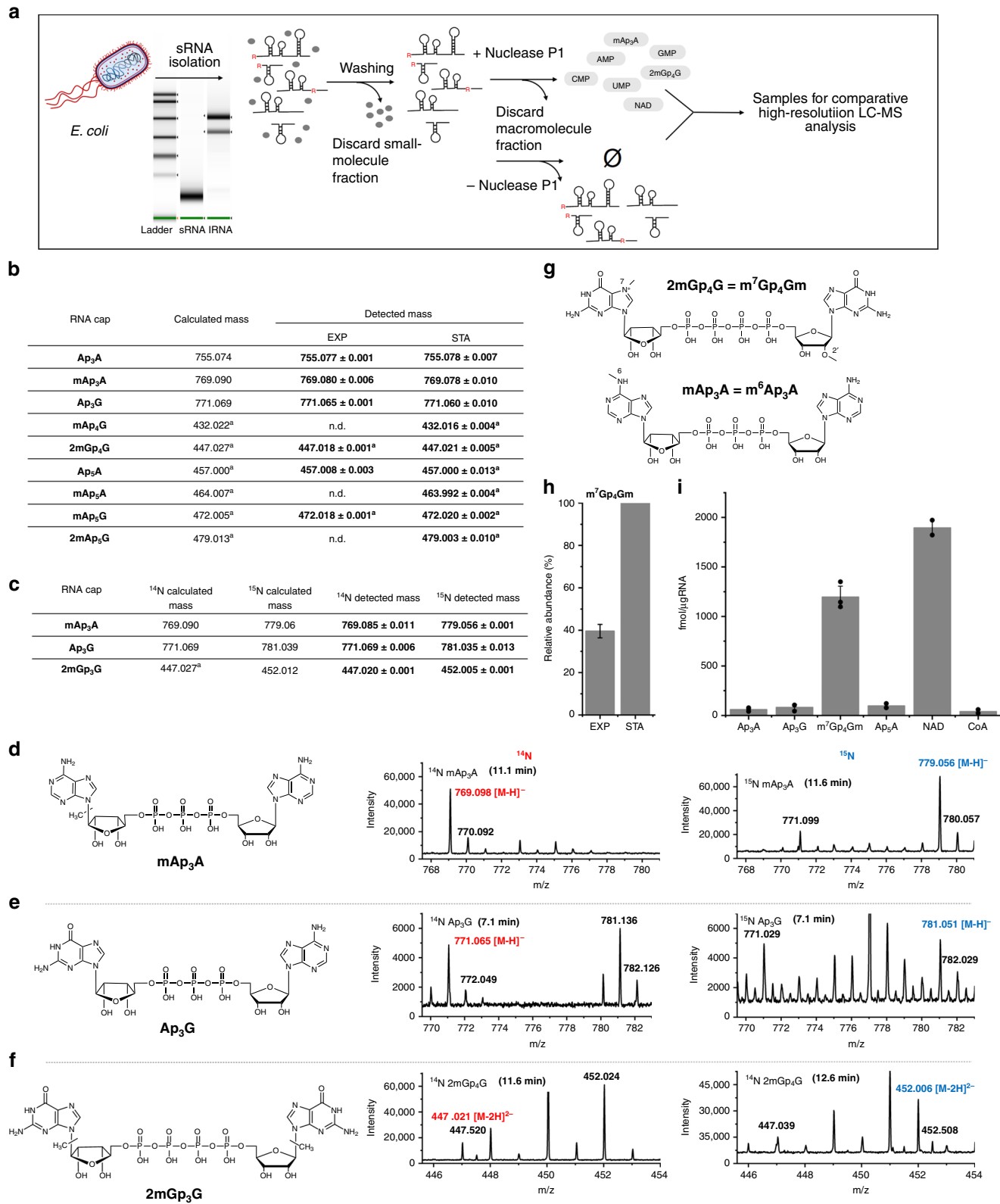

Figure panels a–i

we synthesized all possible mono-methylated Ap$_3$A: m$^6$Ap$_3$A, m$^1$Ap$_3$A, and Amp$_3$A (Supplementary Fig. 9, Supplementary Fig. 20). Based on its retention time, we successfully identified m$^7$Gp$_4$Gm as one of the caps (Fig. 2g, Supplementary Fig. 10). The fragmentation and retention time of the fragmented methylated Ap$_3$A standards (m$^6$Ap$_3$A, m$^1$Ap$_3$A, and Amp$_3$A)

helped us to identify m$^6$Ap$_3$A as another cap in *E. coli* RNA (Fig. 2g, Supplementary Fig. 11).

**Np$_n$N caps quantification.** We compared the amounts of m$^7$Gp$_4$Gm-RNA at various growth stages. The amount of this cap was more than two-fold higher in STA compared to EXP (Fig. 2h).

**Fig. 2 LC–MS detection of naturally occurring NpnN-RNA in E. coli. a** Scheme showing the RNA preparation for comparative LC–MS measurements. Short RNA was isolated from *E. coli* in two growth stages. sRNA was washed from non-covalently bound small molecules by size-exclusion chromatography (SEC) and divided into two parts. One part was treated by Nuclease P1, the other was treated under identical conditions without the addition of Nuclease P1 as negative control. Both samples were subjected to SEC again and the fraction of small molecules was analyzed by LC–MS. **b** Table of detected *m/z* values in LC–MS analysis of digested sRNA from *E. coli* harvested in EXP and STA (all experiments were performed in biological triplicates). **c** Table of detected *m/z* values in LC–MS analysis of digested sRNA from *E. coli* after growth in minimal media with the sole source of nitrogen from $^{14}$NH$_4$Cl ($^{14}$N) or $^{15}$NH$_4$Cl ($^{15}$N) (all experiments were performed in biological triplicates). **d–f** Structures of different RNA caps and MS spectra of the detected *m/z* in RNA from *E. coli* growth in minimal media containing $^{14}$N or $^{15}$N of methyl-Ap$_3$A (**d** *m/z* 769.098 in $^{14}$N, *m/z* 779.056 in $^{15}$N), Ap$_3$G (**e** *m/z* 771.065 in $^{14}$N, *m/z* 781.051 in $^{15}$N), and dimethyl-Gp$_4$G (**f** *m/z* 447.021 in $^{14}$N, *m/z* 452.006 in $^{15}$N). **g** Chemical structure of m$^7$Gp$_4$Gm and m$^6$Ap$_3$A caps detected in *E. coli* RNA. **h** Relative quantification of m$^7$Gp$_4$Gm cap in RNA from EXP and STA growth of *E. coli* (experiments were performed in biological triplicates). **i** Absolute quantification of Ap$_3$A, Ap$_3$G, m$^7$Gp$_4$Gm, Ap$_5$A, NAD, and CoA RNA caps in RNA from STA growth of *E. coli*. All experiments were performed in triplicates and calculated as average values. Error bars indicate standard deviations. Source data are provided in the Source Data file.

In general, a higher number of methylated NpnNs was detected in this phase. This may indicate that the cells in STA lack nutrients and methylate the NpnN-caps to preserve RNA. We also performed an absolute quantification for Ap$_3$A, Ap$_3$G, Ap$_5$A, and m$^7$Gp$_4$Gm, and we compared it with the concentration of known RNA caps, i.e., NAD and CoA in STA (Fig. 2i, Supplementary Fig. 12). While the amount of Ap$_3$A, Ap$_3$G, and Ap$_5$A was comparable with CoA (∽75 fmol per μg of sRNA), the concentration of m$^7$Gp$_4$Gm was significantly higher (1200 fmol per μg of sRNA) and comparable to NAD (1900 fmol per μg of sRNA).

**NpnN-RNA decapping.** Since NpnN-capped RNAs are produced in *E. coli*, degradation mechanisms of the capped RNA in *E. coli* must also exist. ApnA have been reported to be in vitro substrates for the *E. coli* NudiX enzyme NudH (RppH (refs. [12,29])) and ApaH (ref. [30]). RppH is an *E. coli* decapping enzyme of 5′-ppp RNA[7,14,31] and 5′-diphosphate RNA[8]. To assess whether ApnA-capped RNA can be an RppH or ApaH substrate in vivo, we prepared NpnN- and NAD-capped RNA by in vitro transcription, and tested the products as substrates for both enzymes. First, we added RppH or ApaH, to cleave the 5′-capped RNA (Fig. 3a, c) and we analyzed the products by polyacrylamide gel electrophoresis (PAGE; Fig. 3b, d). We then added the terminator to selectively digest 5′-p RNA (Supplementary Fig. 13a, b). Electrophoretic analysis showed that RppH cleaves 5′-ppp RNA and all the NpnN-capped RNA into 5′-p RNA. Ap$_{4-6}$A and Ap$_{4-5}$G-capped RNAs are excellent substrates for RppH in vitro. However, Ap$_3$A-, NAD-RNA, and 5′-ppp RNA are cleaved less efficiently (Fig. 3b). ApaH also efficiently decapped all the NpnN-RNA but left the 5′-ppp RNA intact (Fig. 3d). Electrophoretic analysis of ApaH decapped RNA treated with the terminator showed that Ap$_{3-4}$N-RNA are cleaved into the form of 5′-p RNA. In contrast, the decapping reaction of Ap$_{5-6}$N-RNA lead to 5′-pp RNA or 5′-ppp RNA that cannot be degraded by terminator (Supplementary Fig. 13 b). Because NudC was recently reported as a decapping enzyme of NAD-RNA[1,5], we also tested NpnN-capped RNA as substrates of this enzyme. The only decapping activity was observed for NAD-RNA and NpnN-RNA stayed intact after the NudC treatment (Supplementary Fig. 14).

To understand the substrate specificity of RppH and ApaH, we performed a kinetic study of NpnN-capped RNA. The decapping reaction of Ap$_{4-5}$N RNA by RppH was efficient and within 5 min ~80% of capped RNA was cleaved. While the corresponding 5′-ppp RNA was only decapped by 50% after 40 min (Supplementary Fig. 15). ApaH cleaved all the NpnN-RNA with similar efficiency and only 5′-ppp RNA stayed intact (Supplementary Fig. 16). When we compared the decapping efficiency of both enzymes, we observed the most pronounced differences for 5′-ppp RNA and for NpnN-RNA containing longer phosphate bridge (Fig. 3e).

**NpnN cap methylation role.** To reveal the effect of NpnN-cap methylation on the RNA, we attempted to prepare model

m$^7$Gp$_4$Gm-RNA as a substrate for RppH. Unfortunately, all the methylated forms of Gp$_4$G (i.e., m$^7$Gp$_4$Gm, m$^1$Gp$_4$Gm, and m$^2$Gp$_4$Gm) are poor substrates for the T7 RNAP (Supplementary Fig. 17), which prevents subsequent in vitro studies.

To overcome this problem, we used *E. coli* isolated RNA naturally containing m$^7$Gp$_4$Gm-RNA. We added RppH into the mixture of isolated sRNA with a spiked model Gp$_4$G-RNA to compare the activity of RppH on methylated and non-methylated substrates. We found that the majority of the model Gp$_4$G-RNA was cleaved within 1 h, while the amount of naturally present m$^7$Gp$_4$Gm-cap remained unchanged (Fig. 4a). A similar effect was also observed for the methylated forms of Ap$_5$G (Supplementary Fig. 18). The spiked Ap$_5$G-RNA was cleaved within 1 h, but the amount of the mAp$_5$G-cap decreased by 25 % and the 2mAp$_5$G-cap stayed intact. This confirms that the methylation protects the NpnN-capped RNA from cleavage by RppH.

We also tested ApaH (the same concentration as RppH) on the mixture of isolated sRNA with a spiked model Gp$_4$G-RNA. Surprisingly, the naturally occurring m$^7$Gp$_4$G and model Gp$_4$G caps were completely cleaved within 1 h (Fig. 4a).

To unravel the possible molecular basis for the inertness of RNA with methylated NpnN-caps to RppH cleavage, we performed molecular dynamics (MD) simulations of the interaction between RppH and two models of NpnN-capped RNA (Gp$_4$G-G and m$^7$Gp$_4$Gm-G). In the MD simulations, we observed that the interactions with the arginines R28 and R86 were lost when the methyl groups were present (Fig. 4b). These arginines are responsible for the purines binding via cation-π stacking, which is disrupted by the positive charge introduced to the purine ring by the methylation. These findings demonstrate how the methylations of NpnN-caps in RNA can hamper the decapping by RppH.

## Discussion

In summary, we identified NpnNs as 5′-RNA caps, which are incorporated into RNA by highly divergent RNAPs. We found that NpnN RNA was cleaved by the *E. coli* RppH and ApaH decapping enzymes. Caps with long polyphosphate chains were cleaved the most efficiently and are better substrates for RppH than 5′-ppp RNA, while ApaH did not show any specific selectivity and cleaved all the NpnN caps. The main difference between these enzymes, is the inability of ApaH to cleave 5′-ppp RNA. In the cells, we detected the presence of both methylated and non-methylated NpnNs in the sRNA. We determined the positions of the methyl groups in m$^7$Gp$_4$Gm and m$^6$Ap$_3$A caps. LC–MS experiments revealed that the methylation protects the caps from RppH cleavage but not from ApaH cleavage. This suggests that the cell may regulate the presence of NpnN-RNA by the expression of these two enzymes, as evidenced in *Salmonella typhimurium*[32], where the mRNA levels of ApaH are kept at a three-fold lower level than those of RppH. In the late stationary phase, the levels of both transcripts decrease significantly in comparison

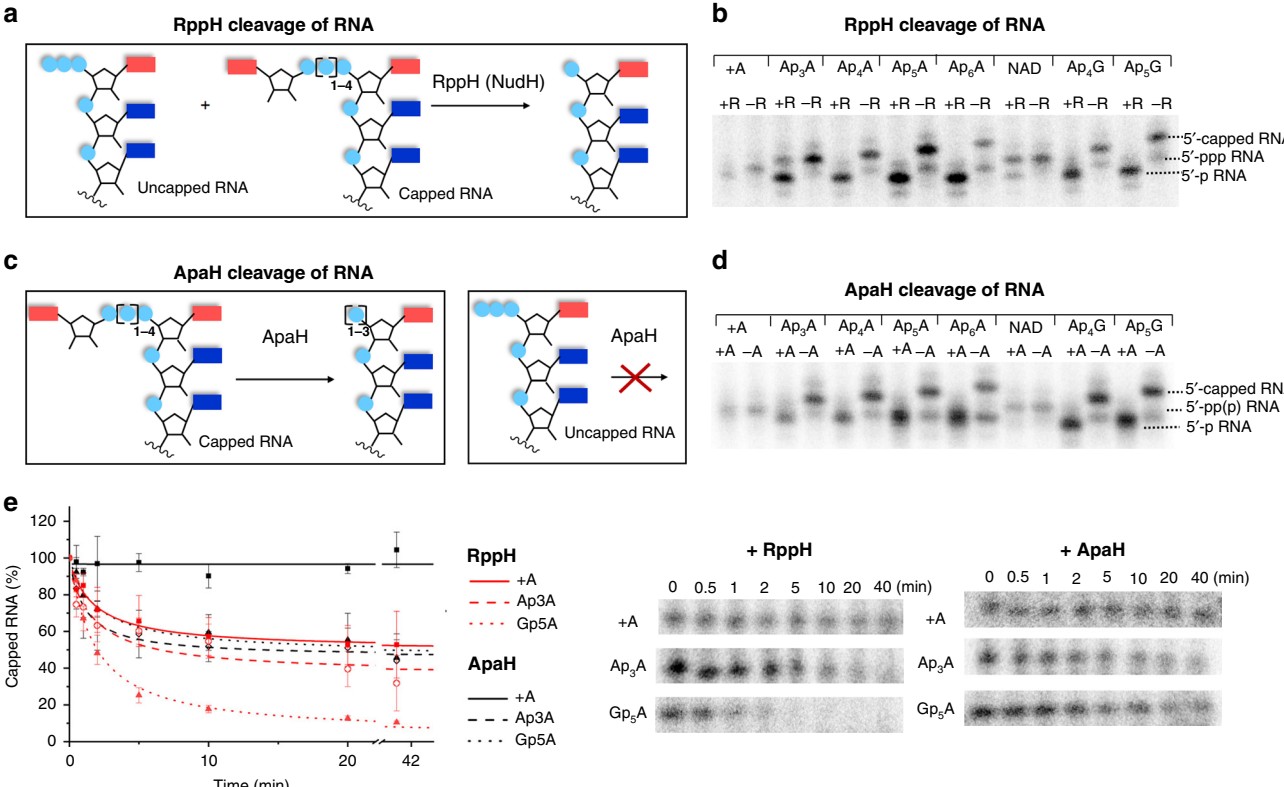

**Fig. 3 RppH and ApaH cleavage of Np$_n$N-capped RNA ($^{32}$P labeled). a** Scheme showing the cleavage of capped and uncapped RNA (in vitro transcribed with T7 RNAP) using RppH. **b** PAGE (12%) analysis of the (α-$^{32}$P GTP labeled) in vitro transcribed RNA (35 nt) treated with RppH (+R) or without RppH (−R; all the experiments were performed in triplicates). **c** Scheme showing the cleavage of capped and uncapped RNA (in vitro transcribed with T7 RNAP) using ApaH. **d** PAGE (12%) analysis of in vitro transcribed RNA (35 nt) treated with ApaH (+A) or without ApaH (−A; all the experiments were performed in triplicates). **e** Kinetic studies of RppH (red curves) and ApaH (black curves) cleavage of 5′-ppp RNA (solid line), Ap$_3$A-RNA (dashed line), and Gp$_5$A-RNA (dotted line; 35 nt) stopped after 30 s, 1, 2, 5, 10, 20, and 40 min and analyzed by PAGE (12%). All experiments were performed in triplicates and calculated as average values. Error bars indicate standard deviations. Source data are provided in the Source Data file.

with the exponential phase (Supplementary Fig. 19). This is in accordance with our finding that the highest amount of methylated Np$_n$N caps is in STA. Hence, we propose that bacteria use methylated caps to stabilize some RNA under stress (Fig. 4c). In exponential phase, the metabolism is at its highest, and the turnover of macromolecules is fast. The methylated caps are therefore not necessary and were detected in low amounts. In contrast, in stationary phase cells lack nutrients and a strategy is needed to conserve these macromolecules. The methylation of the Np$_n$N-caps can be a way to preserve important RNA molecules. When the cell returns to physiological conditions, more ApaH is expressed leading to the degradation of methylated Np$_n$N-RNA. In this way, the cell can get back to the fast turnover of macromolecules. In human cells, it has been shown that the N6-methylation of the first encoded nucleotide (m$^6$Am, m$^6$A) hampers the cleavage of mRNA by NudiX decapping enzyme Dcp2 (ref. [33]). Our proposed mechanism correlates with this finding and suggests that the strategy, through which cells protect their RNA against decapping by methylation, is general and may also be found in higher organisms. Concurrently, a work reporting the existence of Ap$_4$N-RNA caps in *E. coli* was published[34]. However, we did not detect any of the caps reported therein by our LC–MS technique, as the Ap$_4$N-RNA caps were detected under different stress conditions using a different detection technique.

In conclusion, it is intriguing to consider the possibility that many functions of Np$_n$Ns can be explained via their RNA capping potential. Moreover, the 5′-terminal Np$_n$Ns may interact

with a wide range of cellular partners and influence, e.g., cellular response to starvation. In the near future, besides searching for the methyltransferases responsible for the methylation of the Np$_n$N-caps, the key challenge lies in the development of specific techniques to identify the Np$_n$N-capped RNA.

## Methods

**General**. All chemicals were either purchased from Merck or Jena Biosciences and used without further purification. Oligonucleotides were purchased from Generi Biotech. The list of sequences is present in Supplementary Table 1.

Denaturing polyacrylamide gels (PAGE) were visualized by a Typhoon FLA 9500 imaging system.

**In vitro transcription with T7 RNAP**. In vitro transcription was performed using a standard protocol[23] in a 25 μL mixture containing: 80 ng/μL of template DNA (35 A or 35 G), 1 mM UTP, 1 mM CTP, 0.8 mM GTP, or 0.8 mM ATP, respectively, and 0.2 μL α $^{32}$P GTP or α $^{32}$P ATP (activity: 9.25 MBq in 25 μL), respectively, 1.6 mM (0.2–1.6 mM for incorporation efficiency experiments) Np$_n$Ns, 5% dimethyl sulfoxide (DMSO), 0.12% triton X-100, 12 mM dithiothreitol (DTT), 4.8 mM MgCl$_2$ and 1× reaction buffer for T7 RNAP and 62.5 units of T7 RNAP (New England BioLabs, NEB). The mixture was incubated for 2 h at 37 °C. The samples (3 μL) were mixed with 3 μL of 2× RNA loading dye (NEB) and analyzed by 12% PAGE (600 V, 3.5 h).

**DNAse I treatment**. The DNA template was digested by DNAse I to obtain pure RNA. A total of 25 μL of the transcription mixture, 3 μL of 10× reaction buffer for DNAse I (10 mM Tris-HCl, 2.5 mM MgCl$_2$, 0.5 mM CaCl$_2$, pH 7.6 at 25 °C, supplied with the enzyme), and 4 units of DNAse I (NEB) were incubated at 37 °C for 60 min. The enzyme was heat deactivated at 75 °C for 10 min followed by immediate cooling on ice. All samples were purified on RNA mini Quick Spin Columns (Merck) for further use.

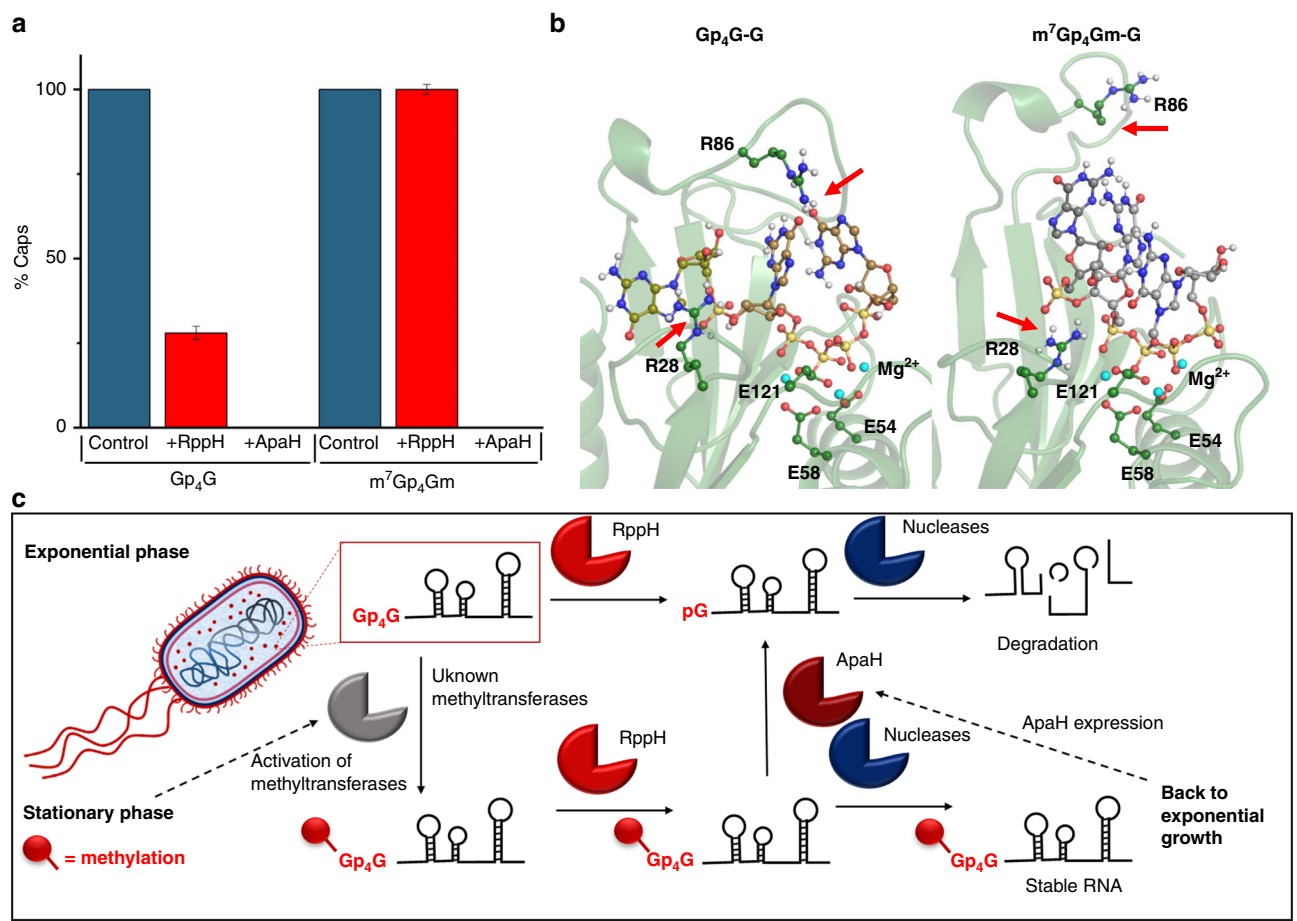

**Fig. 4 Role of RppH in the cleavage of Np$_n$N-capped RNA in *E. coli*. a** Relative abundance of non-methylated Gp$_4$G (left) and m$^7$Gp$_4$Gm (right)-RNA as derived from extracted-ion chromatogram (EIC) in the sRNA fraction spiked with Gp$_4$G-RNA before (blue), after 1 h RppH treatment (red), and after 1 h ApaH treatment (all the experiments were performed in biological triplicates). **b** Snapshots from molecular dynamics simulation of the interaction of RppH with Gp$_4$G-G and m$^7$Gp$_4$Gm-G after 200 ns. **c** Hypothetic cellular processing of RNA in *E. coli* at different stages of growth. Error bars indicate standard deviations. Source data are provided in the Source Data file.

**RNA 5′-polyphosphatase treatment**. A total amount of 2.5–3 µg of RNA (10 µL) were treated with 20 units of 5′-polyphosphatase (Epicenter) in the solution of 1× buffer in a total volume of 20 µL for 1 h at 37 °C. For the negative control, the enzyme was replaced by the same amount of water. The samples (3 µL) were mixed with 3 µL of 2× RNA loading dye and analyzed by 12% PAGE (600 V, 3.5 h).

**Terminator™ 5′-phosphate-dependent exonuclease treatment**. Because of the incompatibility of the buffers, all samples were purified on RNA mini Quick Spin Columns before the reaction. The RNA (500 ng) was treated with 1 unit of Terminator™ 5′-phosphate-dependent exonuclease (Epicenter) in the solution of 1× reaction buffer A and the mixture was incubated at 30 °C for 1 h.

***E. coli* RNAP in vitro transcription**. In vitro transcription was performed in a volume of 40 µL. At first, the mixture contained 10 ng/µL of template plasmid 458 with promoter *rrnB* P1 or 10 ng/µL of the amplicon FV21 possessing *rnaI* promoter, 8 µL of 5× buffer, 90 mM NaCl, 0.2 mM ATP, 0.8 mM Np$_n$Ns, and 1 unit of *E. coli* RNAP holoenzyme (NEB). This mixture was preincubated for 10 min at 37 °C. The transcription was started by addition of an initiation mixture containing 0.2 mM CTP and UTP, 0.15 mM GTP and 0.5 µL α $^{32}$P GTP (activity: 9.25 MBq in 25 µL) and incubated at 37 °C for 1 h.

**Calculations of capped RNA**. The experiments were performed in triplicates and the amount of capped RNA was calculated from PAGE analysis using the software ImageJ[35]. The sum of areas under the peaks corresponding to the 5′-ppp RNA (Ar$_p$) and 5′-capped RNA (Ar$_{cap}$) were integrated. The percentage of 5′-capped RNA species were calculated according to: (Ar$_{cap}$/ Ar$_p$ + Ar$_{cap}$) × 100.

**ApaH—cloning and purification**. The gene for *E. coli* ApaH (EG10048) was prepared by PCR (Expand High Fidelity System, Roche) using primers #3071 and #3072. Subsequently, the gene was inserted into expression vector pET22b

(Novagen) via unique NdeI and XhoI restriction sites. The additional six histidine residues at the C-terminus served as the His-tag facilitating the protein purification process, and the two residues (amino acids L and E) preceding the His-tag, were inserted because of the restriction enzyme (XhoI) used for cloning. The resulting construct (LK2505) was verified by sequencing. The verified plasmid was transformed into expression strain *E. coli* DE3 resulting in strain LK2508.

The expression strain LK2508 was grown in rich lysogeny broth (LB) medium containing ampicillin (100 µg/mL) at 37 °C. Expression of ApaH was induced by IPTG (0.8 mM) when OD$_{600}$ reached 0.6 and the culture was allowed to grow subsequently for 3.5 h at 25 °C. ApaH was purified similarly to RNAP[36]. Briefly, 1 L grown culture was harvested by centrifugation (10 min, 8000 × *g*, 4 °C) and cells were washed with sonication buffer (50 mM Na$_2$HPO$_4$, 300 mM NaCl, 5% glycerol, and 3 mM 2-mercaptoethanol). The cells were then resuspended in 25 mL sonication buffer, sonicated 12 × 10′′ (Hilscher UP200S, Cycle 1, Amplitude 50%) on ice with 1 min intervals between sonications, and subsequently centrifuged (10 min, 26,000 × *g*, 4 °C). The supernatant containing ApaH was mixed with 1 mL Ni-NTA agarose (Qiagen, equilibrated with sonication buffer) and incubated for 2 h on ice (constant mixing). The mixture was then applied on a Poly-Prep chromatography column, washed with 30 mL sonication buffer and then with 30 mL sonication buffer containing 30 mM imidazole. ApaH was eluted with sonication buffer containing 400 mM imidazole (5 × 0.5 mL) and fractions were analyzed by sodium dodecyl sulfate–PAGE. Purified ApaH was dialyzed into storage buffer (50 mM Tris-HCl, pH 8, 100 mM NaCl, 50% glycerol, and 3 mM β-mercaptoethanol) and stored at −20 °C

**NudC[1]**. The expression strain 1827 was grown in rich LB medium containing kanamycin (50 µg/mL) at 37 °C. Expression of NudC was induced by IPTG (0.8 mM) when OD$_{600}$ reached 0.5 and the culture was allowed to grow subsequently for 3 h at 25 °C. The pellet was sonicated 12 × 10′′ (with 50′′ intervals between sonications) in Tris buffer (50 mM Tris-HCl (pH 8), 1 M NaCl, 5 mM MgSO$_4$, 5 mM 2-ME, 5% glycerol, and 5 mM imidazole). The supernatant was mixed with Ni-NTA agarose and washed with 30 mL Tris buffer and 30 mL Tris

buffer with 30 mM imidazole. NudC was eluted with Tris buffer with 400 mM imidazole. Purified NudC was dialyzed into storage buffer (50 mM Tris-HCl (pH 8), 0.5 M NaCl, 50 mM KCl, 1 mM MgCl$_2$, 5 mM DTT, and 50% (v/v) glycerol) and stored at −80 °C.

**Cleavage of capped RNA by RppH, ApaH, or NudC.** To test the cleavage of the 5′-caps, the RNA samples were divided into two parts. The positive control contained ~700 ng of the RNA (in vitro transcription, DNAse I treatment, purified on RNA mini Quick Spin Columns), for RppH cleavage: 2 µL of 10× buffer 2 (500 mM NaCl, 100 mM Tris-HCl, 100 mM MgCl$_2$, 10 mM DTT, pH 7.9 at 25 °C, supplied with the enzyme) and 650 nM RppH (NEB). For ApaH or NudC cleavage: 2 µL of 10× buffer NudC (1 M KCl, 100 mM Tris-HCl, 20 mM MgCl$_2$, 20 mM DTT, pH 7.2 at 25 °C) and 650 nM ApaH or 6.5 µM NudC. RNA was replaced by water for the negative control. The mixtures were incubated at 37 °C for 1 h and purified on RNA mini Quick Spin Columns. A total of 3 µL of the purified samples were mixed with 3 µL of 2× RNA loading dye for analysis and 6 µL were used for the Terminator treatment.

**Kinetic studies.** For the kinetic studies, RNA samples (35 nt) after in vitro transcription with α $^{32}$P GTP, DNase I treatment, and purification on RNA mini Quick Spin Columns were used. The 20 µL mixture contained 2400 ng of the studied RNA, buffer 2, and 65 nM RppH enzyme or buffer NudC and 65 nM ApaH. The mixture was incubated at 37 °C, aliquots of 2 µL were collected at 0, 0.5, 1, 2, 5, 10, 20, and 40 min and mixed immediately with 2 µL of 2× RNA loading dye. All samples were analyzed by 12% PAGE. The experiments were performed in triplicates and the amount of capped RNA was calculated from PAGE analysis using the software ImageJ. The areas under the peaks corresponding to 5′-ppp RNA or 5′-capped RNA were integrated and plotted against time by regarding the area at 0 min as 100%.

***E. coli* growth condition.** The E. coli strain DH5α containing pUC18 was used for studying RNA modification in two different growth conditions: exponential phase and stationary phase. The growths were performed in LB medium (Merck) in the presence of 0.1 mg/mL of Ampicillin (Amp, Merck). Parallel cultures of E. coli were inoculated in Erlenmeyer flasks from cultures grown on LB Agar. Cultures (1 L) were grown at 37 °C until an optical density at 600 nm (OD$_{600}$) of 0.3 was reached (exponential phase, EXP). Cultures (1 L) were harvested after 2 h they reached an OD$_{600}$ of 1.6 (stationary phase, STA). All cells were harvested by centrifugation at 5000 × g for 10 min. The pellets were washed once with PBS and stored at −80 °C.

**$^{14}$N/$^{15}$N isotope RNA labeling.** E. coli strains DH5α containing pUC18 were grown in M9 minimal medium containing 1× M9 salts (5× concentrate: 34 g/L Na$_2$HPO$_4$, 29 g/L of KH$_2$PO$_4$, 2.5 g/L of NaCl, and 5 g/l NH$_4$Cl), 2 mM MgSO$_4$, 0.1 mM CaCl$_2$, and 0.4% glucose). The growth was performed in the presence of 0.1 mg/mL of Amp. Parallel cultures (1 L each) containing $^{14}$NH$_4$Cl or $^{15}$NH$_4$Cl (as sole source of nitrogen) were inoculated in Erlenmeyer flasks from cultures grown overnight in M9 medium (50 mL). All cells were harvested by centrifugation at 5000 × g for 10 min after 48 h of growth (OD = 0.5), washed once with PBS and stored at −80 °C.

**Isolation and purification of sRNA.** sRNA from E. coli was isolated using the RNAzol protocol (Merck). The pellets (stemming from 1 L culture of exponential growth and from 0.5 L stationary phase growth) were suspended in 4 mL lysozyme solution (1 mg/mL, Merck) each and incubated for 1 h at 7 °C. RNAzol (10 mL) was added to the lysate to isolate the RNA. After shaking vigorously for 15 s, the mixture was incubated for 15 min on ice and centrifuged at 6000 × g for 45 min. In this step, the DNA, proteins, and most polysaccharides formed a semisolid pellet at the bottom of the tube. The RNA, which remained in the supernatant, was mixed with 0.4 volumes of 75% ethanol (v/v). After storing on ice for 10 min, the sample was centrifuged at 6000 × g for 16 min for pelleting the long RNA. The supernatant containing sRNA was mixed with 0.8 volumes of isopropanol. After storing on ice for 30 min, the sample was centrifuged at 6000 × g for 40 min for pelleting the sRNA. The pellets containing the long and sRNA were washed twice with 75% ethanol (v/v) and 70% isopropanol (v/v), respectively. Both pellets were dissolved in small amounts of water and stored at −80 °C. To remove possible contaminants, all sRNA samples were submitted to size-exclusion chromatography for five times using an Amicon Ultra 3 KDa cut-off (Merck). All experiments were performed in at least triplicates.

**RppH and ApaH experiments with sRNA.** sRNA (1 mg) from stationary phase was spiked with 10 µg of model Gp$_4$G-RNA or Ap$_5$G-RNA and divided into two halves. The first part was diluted to a final volume of 300 µL with 30 µL of 10 × buffer 2 (for RppH) or with 10 × buffer NudC (for ApaH) and RppH (NEB), or ApaH were added to reach the final concentration of 5 µ. The second half was used as a negative control and also diluted to a final volume of 300 µL using buffer without the addition of RppH. The mixtures were incubated at 37 °C for 1 h, and further used for LC–MS analysis.

**sRNA digestion for LC–MS.** A total amount of 1 mg of sRNA was divided into two aliquots; one aliquot of 0.5 mg was digested by 10 U of Nuclease P1 (Merck) in 50 mM ammonium acetate buffer (pH 4.5) at 37 °C for 1 h. The second aliquot of 0.5 mg was incubated without enzyme and used as negative control. The digest was purified over Amicon-Millipore filters 10 kDa (Merck). The flow through was dried up on a Speedvac system and dissolved in 10 µL of a mixture of acetonitrile (10%) and ammonium acetate (10 mM, pH 12). The final pH was adjusted to 10 using a solution of NaOH.

**In vitro transcription with T7 RNAP for synthesis of pppGpG.** In vitro transcription was performed in the 50 µL mixture containing 120 ng/µL of template DNA (2ntG), 2.0 mM GTP, 5% DMSO, 0.12% triton X-100, 12 mM DTT, 4.8 mM magnesium chloride, and 1 × reaction buffer for T7 RNAP and 125 units of T7 RNAP. The mixture was incubated for 2 h at 37 °C. The mixture was filtered on an Amicon Ultra 0.5 Centrifugal Filter 3 kDa (Merck), which was then washed with water (2 × 150 µL). The combined flow through was evaporated on a Speedvac system.

**Synthesis of $N^1$-methyladenosine (m$^1$A)[37,38].** Iodomethane (1.87 mL, 30 mmol) was added dropwise to a suspension of adenosine (2 g, 7.5 mmol) in N,N-dimethylacetamide (20 mL). The reaction mixture was stirred overnight at room temperature (RT). Celite (200 mg) was added and the resulting suspension was stirred at RT for 30 min. Afterward, the suspension was filtered and the solids were washed with dry acetone (100 mL). Product precipitated at 5 °C. After filtration and washing with cold acetone, the product was isolated as an off-white powder in 81% yield (1.7 g, 6.0 mmol). $^1$H nuclear magnetic resonance (NMR) (401 MHz, D$_2$O) δ 8.56 (s, 1 H), 8.55 (s, 1 H), 6.15 (d, J = 5.2 Hz, 1 H), 4.81 (1 H, overlapped with HDO), 4.46 (dd, J = 5.3, 4.3 Hz, 1 H), 4.28 (td, J = 4.4, 3.0 Hz, 1 H), 4.02 − 3.80 (m, 5 H) ppm. $^{13}$C NMR (101 MHz, D$_2$O) δ 151.73, 148.49, 147.56, 143.87, 120.05, 89.21, 86.15, 74.83, 70.80, 61.75, 38.51 ppm. MS (ESI$^+$) m/z (%): 282 (100), 304 (10, +Na). High resolution mass spectrometry (HRMS) (ESI) m/z: [(M)$^+$] (C$_{11}$H$_{16}$O$_4$N$_5$) calc.: 282.11940, found: 282.11968.

**Synthesis of $N^6$-methyladenosine (m$^6$A)[39].** $N^1$-methyladenosine (1.0 g, 3.5 mmol) was suspended in an aqueous solution of NaOH (0.25 M, 50 mL) and heated at 110 °C for 75 min. Then, the solution was neutralized by addition of 10% water solution of p-toluenesulfonic acid. Then, water was removed under reduced pressure (maximum bath temperature was always 40 °C). Methanol was added to the solid residue and resulting suspension was heated at 80 °C for 5 min. Methanol was removed under reduced pressure. Finally, the solid residue was suspended in EtOAc and the resulting suspension was refluxed at 80 °C overnight. After filtration, EtOAc from the liquors was removed on rotary evaporator. Solids were dried under vacuum to get the $N^6$-methyladenosine in 50% yield (500 mg, 1.8 mmol). $^1$H NMR (400 MHz, D$_2$O) δ 8.24 (s, 1 H), 8.19 (s, 1 H), 6.03 (d, J = 6.2 Hz, 1 H), 4.78 − 4.77 (m, 1 H), 4.46 − 4.37 (m, 1 H), 4.29 (d, J = 3.1 Hz, 1 H), 3.97 − 3.80 (m, 2 H), and 3.07 (s, 3 H) ppm. $^{13}$C NMR (101 MHz, D$_2$O) δ 154.72, 152.16, 151.50, 139.76, 125.86, 88.27, 85.77, 73.69, 70.62, 61.50, 29.76 ppm. MS (ESI$^+$) m/z (%): 282 (100), 304 (90, +Na). HRMS (ESI) m/z: [(M)$^+$] (C$_{11}$H$_{16}$ O$_4$N$_5$) calc.: 282.11968, found: 282.11938.

**Monophosphorylation of m$^1$A, m$^6$A and 2′-O-methyladenosine.** The respective methyladenosine derivative (1 eq.) was suspended in trimethyl phosphate (0.1 M, final concentration). Resulting suspension was cooled at 0 °C and POCl$_3$ (2.0 eq.) was added dropwise. Reaction mixture was stirred at 0 °C until complete consumption of the starting material according high-performance liquid chromatography (HPLC). Then, water was added and the resulting solution was neutralized with NaOH 6 M and HCl 1 M. The respective methyladenosine monophosphates were isolated after preparative HPLC (A—triethylammonium acetate 0.1 M, pH 7.0, B—acetonitrile). Then, water from each fraction was removed under reduced pressure. Final compounds were obtained after lyophilisation.

$N^1$-methyladenosine 5′-monophosphate (yield 63%). $^1$H NMR (401 MHz, D$_2$O) δ 8.63 (s, 1 H), 8.52 (s, 1 H), 6.17 (d, J = 5.4 Hz, 1 H), 4.77 (1 H, overlapped with HDO), 4.51 (dd, J = 5.1, 3.8 Hz, 1 H), 4.39 (dq, J = 5.3, 3.0 Hz, 1 H), 4.22 − 4.06 (m, 2 H), 3.93 (s, 3 H) ppm. $^{13}$C NMR (101 MHz, D$_2$O) δ 151.03, 147.95, 147.16, 142.82, 118.98, 88.01, 84.52, 74.76, 70.44, 64.35, 37.76, 8.28 ppm. MS (ESI$^+$) m/z (%): 362 (100), 384 (25, +Na). HRMS (ESI) m/z: [(M)$^+$] (C$_{11}$H$_{17}$ O$_7$N$_5$P) calc.: 362.08601, found: 362.08607.

$N^6$-methyladenosine 5′-monophosphate (yield 31%)[39]. $^1$H NMR (401 MHz, D$_2$O) δ 8.25 (s, 1 H), 7.99 (s, 1 H), 5.94 (d, J = 5.7 Hz, 1 H), 4.60 (t, J = 5.4 Hz, 1 H), 4.38 (dd, J = 5.2, 3.7 Hz, 1 H), 4.27 (dd, J = 3.6, 2.2 Hz, 1 H), 4.03 (td, J = 5.0, 3.0 Hz, 2 H), 2.94 − 2.85 (m, 3 H) ppm. $^{13}$C NMR (101 MHz, D$_2$O) δ 154.30, 152.04, 150.72, 139.16, 118.50, 87.00, 84.05, 83.96, 70.46, 64.48, 29.66 ppm. MS (ESI$^−$) m/z (%): 360 (100), 382 (10, +Na). HRMS (ESI) m/z: [(M)$^−$] (C$_{11}$H$_{15}$ O$_7$N$_5$P) calc.: 360.07146, found: 360.07108.

2′-O-methyladenosine 5′-monophosphate (yield 47%)[40]. $^1$H NMR (401 MHz, D$_2$O) δ 8.39 (s, 1 H), 8.11 (s, 1 H), 6.07 (d, J = 5.9 Hz, 1 H), 4.56 (dd, J = 5.1, 3.4 Hz, 1 H), 4.37 (dd, J = 6.0, 5.0 Hz, 1 H), 4.28 (dd, J = 3.4, 2.3 Hz, 1 H), 4.03 (td, J = 3.0, 1.3 Hz, 2 H), 3.36 (s, 3 H) ppm. $^{13}$C NMR (101 MHz, D$_2$O) δ 155.09, 152.22, 148.87, 140.07, 118.51, 85.47, 84.72, 83.26, 69.13, 64.46, 58.22 ppm. MS

(ESI⁻) *m/z* (%): 360 (100). HRMS (ESI) *m/z*: [(M)⁻] ($C_{11}H_{15}O_7N_5P$) calc.: 360.07146, found: 360.07077.

**Synthesis of adenosine 5′-diphosphoimidazolide**. Adenosine 5′-diphosphate (0.5 g, 1.17 mmol), imidazole (0.64 g, 9.4 mmol), and 2,2′-dithiopyridine (0.77 g, 3.5 mmol) were suspended in dimethylformamide (DMF; 10 mL). Then, tri-methylamine (323 μL, 2.3 mmol) and triphenylphosphine (0.92 g, 3.5 mmol) were added. Resulting suspension was stirred at RT for 24 h. The resulting clear solution was poured into a flask containing anhydrous sodium perchlorate (1 eq.) dissolved in dry and cold acetone (8 mL per 1.0 mL of DMF). After cooling at 4 °C for 30 min, liquids were separated from solids by centrifugation. Resulting solid pellet was washed four times with dry and cold acetone, and was centrifuged each time. Finally, solids were washed with diethyl ether and were centrifuged. Product was isolated as an off-white solid in 45% yield (0.25 g, 0.52 mmol). Solids were dried overnight under vacuum and used in the next step without further purification. MS (ESI⁺) *m/z* (%): 500 (25, +Na), 522 (100, +2Na), 544 (40, +3Na). HRMS (ESI) *m/z*: [M+Na⁺] ($C_{13}H_{17}O_9N_7NaP_2$) calc.: 500.04552, found: 500.04558.

**General synthesis of mAp₃A standards**. Respective methyladenosine 5′-mono-phosphate (1.2 eq.), adenosine 5′-diphosphoimidazolide (1.0 eq.), and dry MgCl₂ (3 eq.) were suspended in dry DMF. Resulting suspension was stirred at RT for 6 h. The reaction was stopped by addition of H₂O. Products were isolated after pre-parative HPLC (A—triethylammonium acetate 0.1 M, pH 7, B—acetonitrile). Co-distillations with water followed by several freeze-drying from water gave off-white solid products.

**$P^1$-[5′-($N^1$-methyladenosyl)] $P^3$-(5′-adenosyl) triphosphate (yield 40%)**. ¹H NMR (500 MHz, D₂O, internal ref. *t*BuOH, 1.24 ppm) δ 8.50 (s, 1 H), 8.45 (d, *J* = 2.5 Hz, 1 H), 8.31 (s, 1 H), 8.15 (s, 1 H), 6.08 (d, *J* = 5.1 Hz, 1 H), 6.03 (d, *J* = 5.4 Hz, 1 H), 4.74 (t, *J* = 5.0 Hz, 1 H), 4.70 (t, *J* = 5.2 Hz, 1 H), 4.53 (dt, *J* = 9.3, 4.5 Hz, 2 H), 4.39 – 4.21 (m, 6 H), 3.86 (s, 3 H) ppm. ¹³C NMR (125.7 MHz, D₂O, internal ref. *t*BuOH, 32.43 ppm) δ 157.91, 155.55, 153.17, 151.45, 150.71, 149.42, 145.38, 142.31, 121.06, 120.73, 90.68, 89.61, 86.79 – 86.72 (d, *J* = 9.1 Hz, C₄′), 86.42 – 86.34 (d, *J* = 9.1 Hz, C₄′), 77.83, 77.54, 72.94, 72.91, 67.85 – 67.81 (d, *J* = 5.3 Hz, C₅′), 67.69 – 67.65 (d, *J* = 5.3 Hz, C₅′), 40.54 ppm. ³¹P NMR (202.4 MHz, D₂O, external ref. H₃PO₄, 0.0 ppm) δ −10.41 (dd, *J* = 19.4, 11.4 Hz), −21.89 (t, *J* = 19.4 Hz) ppm. MS (ESI⁺) *m/z* (%): 771 (100). HRMS (ESI) *m/z*: [M⁺] ($C_{21}H_{30}O_{16}N_{10}P_3$) calc.: 771.10486, found: 771.10593.

**$P^1$-[5′-($N^6$-methyladenosyl)] $P^3$-(5′-adenosyl) triphosphate (yield 30%)**. ¹H NMR (500 MHz, D₂O, internal ref. *t*BuOH, 1.24 ppm) δ 8.26 (s, 1 H), 8.21 (s, 1 H), 8.09 (m, *J* = 2.3 Hz, 2 H), 6.01 (d, *J* = 4.7 Hz, 1 H), 5.99 (d, *J* = 4.5 Hz, 1 H), 4.63 (t, *J* = 4.7 Hz, 1 H), 4.59 (t, *J* = 4.6 Hz, 1 H), 4.49 (d, *J* = 13.1 Hz, 1 H), 4.37 – 4.30 (m, 5 H), 4.27 (m, 2 H), 2.97 (bs, 3 H) ppm. ¹³C NMR (125.7 MHz, D₂O, internal ref. *t*BuOH, 32.43 ppm) δ 157.66, 157.06, 155.39, 155.07, 150.90, 141.74, 141.21, 120.63, 90.13, 90.02, 85.91 – 85.83 (d, *J* = 9.1 Hz, C₄′), 85.79 – 85.71 (d, *J* = 9.1 Hz, C₄′), 77.85, 77.81, 72.50, 72.36, 67.49 – 67.44 (d, *J* = 5.2 Hz, C₅′), 67.39 – 67.35 (d, *J* = 5.3 Hz, C₅′), 25.86 ppm. ³¹P NMR (202.4 MHz, D₂O, external ref. H₃PO₄, 0.0 ppm) δ −10.42 (dd, *J* = 19.4, 5.3 Hz), −21.89 (t, *J* = 19.4 Hz) ppm. MS (ESI⁻) *m/z* (%): 769 (100), 791 (80, +Na), 813 (20, +2Na). HRMS (ESI) *m/z*: [M⁻] ($C_{21}H_{28}O_{16}N_{10}P_3$) calc.: 769.09031, found: 769.08953.

**$P^1$-[5′-(2′-$O$-methyladenosyl)] $P^3$-(5′-adenosyl) triphosphate (yield 50%)**. ¹H NMR (500 MHz, D₂O, internal ref. *t*BuOH, 1.24 ppm) δ 8.30 (s, 1 H), 8.28 (s, 1 H), 8.09 (s, 1 H), 8.09 (s, 1 H), 6.03 (d, *J* = 4.6 Hz, 1 H), 5.97 (d, *J* = 5.2 Hz, 1 H), 4.59 (t, *J* = 5.1 Hz, 1 H), 4.57 (t, *J* = 4.7 Hz, 1 H), 4.48 (t, *J* = 4.4 Hz, 1 H), 4.35 – 4.24 (m, 6 H), 4.21 (t, *J* = 4.8 Hz, 1 H), 3.50 (s, 3 H) ppm. ¹³C NMR (125.7 MHz, D₂O, internal ref. *t*BuOH, 32.43 ppm) δ 157.89, 157.86, 155.40, 155.34, 151.24, 150.98, 142.18, 142.10, 120.88, 89.85, 88.27, 86.44, 86.26 – 86.21 (d, *J* = 9.1 Hz, C₄′), 86.20 − 86.16 (d, *J* = 9.1 Hz, C₄′), 77.60, 72.80, 71.08, 67.77 – 67.73 (d, *J* = 5.3 Hz, C₅′), 67.39 – 67.36 (d, *J* = 5.2 Hz, C₅′), 61.13, 32.43 ppm. ³¹P NMR (202.4 MHz, D₂O, external ref. H₃PO₄, 0.0 ppm) δ −10.38 (dd, *J* = 19.4, 5.8 Hz), −21.93 (t, *J* = 19.4 Hz) ppm. MS (ESI⁻) *m/z* (%): 769 (100), 791 (20, +Na), 813 (5, +2Na). HRMS (ESI) *m/z*: [M⁻] ($C_{21}H_{28}O_{16}N_{10}P_3$) calc.: 769.09031, found: 769.08968.

**LC–MS data collection and analysis**. LC–MS was performed using a Waters Acquity UPLC SYNAPT G2 instrument with an Acquity UPLC BEH Amide column (1.7 μm, 2.1 mm × 150 mm, Waters). The mobile phase A consisted of 10 mM ammonium acetate, pH 9, and the mobile phase B of 100% acetonitrile. The flow rate was kept at 0.25 mL/min and the mobile phase composition gradient was as follows: 80% B for 2 min; linear decrease to 68.7% B over 13 min; linear decrease to 5% B over 3 min; maintaining 5% B for 2 min; returning linearly to 80% B over 2 min. For the analysis, electrospray ionization (ESI) was used with a capillary voltage of 1.80 kV, a sampling cone voltage of 20.0 V, and an extraction cone voltage of 4.0 V. The source temperature was 120 °C and the desolvation temperature 550 °C, the cone gas flow

rate was 50 L/h and the desolvation gas flow rate 250 L/h. The detector was operated in negative ion mode. For each sample, 8 μL of the dissolved material was injected.

Triplicates of Nuclease P1-digested RNA samples were used to identify Np$_n$Ns. Ions with <50 counts were not considered for further analysis. MassLynx software was used for data analysis and quantification of the relative abundance of dimethyl-Gp₄G.

**MS³ fragmentation analysis**. The fragmentations studies were performed using SCIEX QTRAP 6500+ instrument with an Acquity UPLC BEH Amide column (1.7 μm, 2.1 mm × 150 mm, Waters). Mobile phase A was 10 mM ammonium acetate pH 9, and mobile phase B was 100 % acetonitrile.

*Ap₃A*: For the determination of Ap₃A structure the flow rate was a constant 0.25 mL/min and the mobile phase composition was as follows: 80% B for 2 min; linear decrease over 3 min to 50% B; and maintain at 50% B for 1 min before returning linearly to 80% B over 2 min. ESI was used with curtain gas of 35 (arbitrary units), Ionspray voltage of 4.5 kV. The ion source gas was 50 (arbitrary units), and the drying gas temperature was 400 °C. The declustering potential was −200 V, the entrance potential −10 V, collision energy −46 V, excitation energy 0.1 V. The detector was operated in negative ion mode. For the confirmation of Ap₃A structure, the first precursor ion was [M − H]⁻ at *m/z* 754.96 and as second ion the ATP fragment was selected [M − H]⁻ at *m/z* 487.8. For each sample, 8 μL of the dissolved material was injected.

*mAp₃A*: For the determination of mAp3A structure the flow rate was a constant 0.20 mL/min and the mobile phase composition was as follows: 80% B for 2 min; linear decrease over 12 min to 50% B; and maintain at 50% B for 1 min before returning linearly to 80% B over 2 min. ESI was used with curtain gas of 20 (arbitrary units), ionspray voltage of 4.5 kV. The ion source gas was 50 (arbitrary units), and the drying gas temperature was 400 °C. The declustering potential was −300 V, the entrance potential −10 V, collision energy −54 V, and excitation energy 0.1 V. The detector was operated in negative ion mode. For the identification of mAp₃A structure, the first precursor ion was [M − H]⁻ at *m/z* 768.908 and as second ion the dehydrated mAMP fragment was selected [M − H]⁻ at *m/z* 341.8. For each sample, 8 μL of the dissolved material was injected.

**Quantification**. A total of 2 mg of sRNA from STA was divided in four aliquots. After digestion with NuP1, each aliquot was spiked with increasing concentration of Np$_n$Ns (Ap₃A, Ap₃G, m⁷Gp₄G, and Ap₅A), NAD, and CoA. After LC–MS analysis of all the aliquots the area under the peak was calculated and plotted against the concentration. The experimental points were fitted with a liner regression and the intercept with *X*-axis represent the concentration of the cap in the sample. The quantification was repeated three times and the final value was obtained from the average of the three measurement with relative SD.

**MD simulations**. All models were based on a crystal structure of *E. coli* RppH in complex with modified ppcpAG 5′-capped dinucleotide (PDB ID 4S2Y). We modified the adenine to guanine and extended the 5′-nucleotide triphosphate into Gp₄G or the methylated m⁷Gp₄Gm. Hydrogens were added so that the amino acids were present in the standard protonation state at pH 7. Three catalytic magnesium ions were kept in place. The complex was solvated in an 83 × 68 × 71 Å box of water molecules. Some water molecules were replaced by sodium and chlorine ions to neutralize the complex charge and mimic the cel-lular ionic strength (0.15 M). The MD simulations were performed using the NAMD software[41] and AMBER force field with ff14SB (ref. [42]) parameters for the protein, OL3 (ref. [43]) parameters for RNA, the SPC/E model of water[44], and corresponding parameters for monovalent ions[45] and magnesium[46]. Parameters for non-canonical RNA caps were constructed using existing RNA parameters and available parameters for polyphosphates[47]. Partial atomic charges were fitted based on quantum-chemical HF/6-31 G∗ calculations using Gaussian 09 (ref. [48]) using the RESP procedure AmberTools[49]. We used periodic boundary condi-tions in the MD simulations to emulate a bulk solvent. An isobaric-isothermal ensemble (NPT) scheme with Langevain temperature and pressure control was applied. The following equilibration protocol was used: (1) 1000 steps of con-jugate gradient minimization with restraints on heavy atoms of the protein and RNA, (2) heating from 0 to 310 K followed by 10 ps of MD with above restraints, (3) 1000 steps of minimization without restraints, and (4) heating from 0 to 310 K followed by 100 ps of unrestrained equilibration. We used 1 fs MD integration steps for the equilibration phase and 2 fs for the 200 ns production phase. The final snapshots were visualized using PyMol[50].

**Reporting summary**. Further information on research design is available in the Nature Research Reporting Summary linked to this article.

## Data availability

A reporting summary for this Article is available as a Supplementary Information file. Source data used in this work are publicly available at https://figshare.com/articles/Source_Data_Hudecek_Benoni_Cahova_2019_xlsx/11637132. All other data are available from the authors upon reasonable request.

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

## Acknowledgements

Jiří Potužník, Maria-Bianca Mititelu, Victoria Fincke, Carlos Murillo Almuzara and other members of Cahova lab for their help and discussion. We acknowledge funding from the Ministry of Education, Youth, and Sports (Czech Republic), program ERC CZ (LL1603), and Czech Science Foundation (17-03419S; to LK).

## Author contributions

O.H., R.B., and H.C. designed the experiments and coordinated the project. O.H. and R.B. performed the experiments. P.E.R-.G. synthesized mAp₃A. H.Š. prepared ApaH. M.C. performed the MD study. M.H. and J. C. consulted the LC–MS results. J.C., L.R., L.K., and H.C. supervised the work. O.H., R.B., M.C., and H.C. wrote the paper.

## Competing interests

The authors declare no competing interests.
