## [Peer Review File · Nature Communications]

Reviewers' comments:

Reviewer #1 (Remarks to the Author):

Using LC-MS, Hudecek et al. provide evidence for the presence of both methylated and non-methylated short NpnN-capped RNAs in E.coli cells. They further demonstrate that RppH (E.coli 5' pyrophosphohydrolase) efficiently cleaves dinucleoside polyphosphates from the RNA and its activity is inhibited by NpnN methylation. Interestingly, the authors found that amounts of NpnNs and their methylation increased in the stationary phase of bacteria growth (stress condition) and propose that bacteria use methylated caps to stabilize some RNAs under stress. The manuscript presents interesting findings, however the framework of this manuscript needs to be substantially improved.

Specific comments:

1. The authors detected Ap3A, Ap3G and Ap5A as well as methyl-Ap3A, dimethyl-Gp4G and methyl-Ap5G caps in exponential phase and additionally methyl-Ap4G, methyl-Ap5A and dimethyl-Ap5G in the stationary phase by LC-MS methodology. It is crucial to identify where the methylation sites are within the cap structure. The entire manuscript is premised on the first nucleotide being the methylation sites, including the proposed resistance to RppH cleavage. The authors could use chemically synthesized standards and LC-MS/MS method to identify and quantify methyl-modified dinucleotide polyphosphates.
2. How abundant are NpnN-capped (and methylated) RNAs? What are their levels compared to RNAs capped with other non-canonical initiating nucleotides, for instance NAD or just uncapped triphosphate RNA? What is the relative ratios of the different dinucleotide caps?
3. Figure 4a should be removed. This is a simulation that may or may not have any relevance to the actual structure. Extensive mutagenesis should be carried out to validate the conserved contacts with corresponding *in vitro* functional studies before this figure is presented in the manuscript.
4. Why did the authors focus only on short RNAs? Does it mean that no NpnNs were not detected in the long RNA pool?
5. page 1, the authors made a point that T7 RNAP can efficiently incorporate NpnN during *in vitro* transcription. It has previously been reported that Gp4G, Gp5G and their methylated counterparts can be efficiently utilized by T7 polymerase (Jemielity et al. RNA 2003). This should be cited.
6. Page 2, the extended data Fig 1c shows 33% capping for the ATP/NAD = 1 which is comparable to NpnN. If this statement is true, it does not support the sentence "the NAD-capped RNA was produced in lower amounts compared to the majority of NpnNs".

7. Page 5, Figure legend 3, it would be beneficial to include the length of RNA, how the RNA was labeled, and a % of page gel used in the experiment. Additionally, in panel d and e, it should be indicated that exonuclease was present in the reaction as well.

8. The authors should check spelling and words used throughout the manuscript. For example: page 5 “we added RppH to transform the 5'-capped RNA into 5'-p RNA..” Not sure if this would be considered a transformation.

“The decapping reaction of Ap4,5N RNA was almost quantitative ..” what does almost quantitative mean?

Reviewer #2 (Remarks to the Author):

The manuscript by Hudeček et al. provides evidence for the presence of a previously unknown 5' end modifications on RNA isolated from *E. coli* cells. In particular, evidence is presented that small RNAs isolated from *E. coli* contain 5' end dinucleoside polyphosphates (some of which also contain one or two methyl groups). To detect these 5' end structures, the authors use LC-MS methods previously used by David Liu's group in work published in 2009 to detect 5'-NAD caps and 5'-CoA caps on RNAs isolated from *E. coli*. In addition to the LC-MS results, the authors provide two additional pieces of information. First, experiments that show both *E. coli* RNA polymerase and T7 RNA polymerase can use dinucleoside polyphosphates as substrates for transcription initiation in vitro (thus providing a mechanistic explanation for the presence of 5'-end dinucleoside polyphosphate structures on RNAs isolated from cells). Second, in vitro experiments that show *E. coli* RppH can process RNAs containing a 5'-end dinucleoside polyphosphates, leaving a 5' monophosphate.

In principle, the finding that 5'-end dinucleoside polyphosphate “caps” that resemble the structure of the eukaryotic m7G cap are present on RNAs isolated from *E. coli* is suitable for publication in a top journal. However, the current version of the manuscript needs substantial revisions before being appropriate for publication.

Specific changes requested:

(1.) The thesis work should be cited and/or discussed: “Stress-induced capping of mRNA in bacteria” by Daniel Luciano from Joel Belasco's lab (NYU, 2014). The results presented in this thesis suggest particular full-length mRNAs that contain 5' dinucleoside polyphosphate “caps” and suggest the involvement of ApaH rather than RppH in the processing of these cap structures.

(2.) Claims regarding the relative efficiencies of dinucleoside polyphosphates vs. NTPs as initiating nucleotides should be removed from the manuscript unless the appropriate kinetic competition assays are performed. The claim that “NpnNs are superior initiating substrates compared to NAD,” should be removed from the manuscript unless the appropriate kinetic competition assays are performed. Furthermore, NAD-capping (and presumably use of NpnNs as initiating nucleotides) exhibits strong promoter sequence specificity. Thus, any claim regarding efficiency of NpnNs vs. NAD must take into consideration the promoter templates used for the comparison. Analysis of relative efficiencies may also require use of A-less or G-less templates depending upon the NpnN analyzed.

(3.) The authors analyze removal of dinucleoside polyphosphate caps by RppH. Authors should comment on why they did not test NudC (previously shown to remove various adenosine-containing RNA-caps such as NAD), ApaH (which is proposed to remove dinucleoside polyphosphate caps in the work reported in the Luciano lab thesis), or other Nudix-family members in *E. coli*. Without analysis of these other enzymes, it is unclear whether RppH’s activity in vitro reflects a role in the removal of dinucleoside polyphosphate caps in vivo (especially given no differences in dinucleoside polyphosphate caps are observed by LC-MS analysis when RppH is removed from cells). In addition, both unmethylated and methylated caps should be analyzed for sensitivity to NudC and ApaH.

(4.) Many changes in Figures and legends should be made. Where appropriate, figure legends should indicate the number of technical replicates and/or number of biological replicates, and indicate any statistical analysis used. Figure 1e is very confusing and the graph format and color choices make the data essentially uninterpretable. The table in figure 2b should be split into separate tables to make the data that is presented clear. For figures 3b and c, state which RNA polymerase, template, and NTPs were used to generate the RNA products.

(5.) The authors should justify the choice of the *rrnB* P1 promoter for the in vitro transcription experiments using *E. coli* RNAP. *rrnBP1* is an atypical promoter in terms of its biochemical properties (unusual start site, unstable open complexes). Ideally, the authors would provide experiments analyzing capping on at least one standard promoter (such as *lacUV5*). The authors should also provide sequence information for the promoter templates analyzed in vitro in the appropriate figures. Also, it was unclear how experiments using T7 RNAP fit into this story given that T7 RNAP is not present in *E. coli* cells from which these caps are detected.

(6.) For Figure 3c, The RNA products in the second and third lanes from the left of the gel do not show the expected migration as observed in panel b. Have the authors mislabeled the gel? The authors should address why in Figure 3c it appears the RNA product generated in the lane containing NAD⁺ does not appear to carry an NAD-cap capped RNA. Samples in Figure 3 were DNase I treated

using a protocol that involves heat inactivation. NAD is a heat-labile, therefore results obtained with NAD may have been affected by inclusion of a high temperature step.

(7.) For Figure 4c, the authors should better describe and provide a sufficient justification for the inclusion of this molecular dynamics modeling. It was not clear precisely how the authors assigned the location of the methyl groups within NpnN caps. Also, the structure shown in the figure panel is uninformative as the color scheme used makes it difficult to see features of the structure (Mg ions are the same color as amino acids and Gp4G; the orange and red colors are very similar as well).

(8) The authors mention a heat shock growth condition in the method section that is not mentioned elsewhere in the main body of the paper.

(9) Authors should comment on whether any of the signals derived from 5' end modifications of unknown origin presented in the supplemental data of the prior 2009 work from Liu and co-workers correspond to the 5' ends detected in this work.

(10) The authors should comment on why the LC-MS analysis presented in the paper was performed using RNA from DH5alpha cells, but the rppH knockout strain analyzed was generated in a different strain. Why not analyze DH5alpha cells that contain the rppH deletion?

(11) For Figure 3, panels d and e, it may be difficult to analyze the effect of RppH on ppp-RNA due to the difficulty in separating ppp-RNA and p-RNA using the electrophoresis conditions used.

Reviewer #3 (Remarks to the Author):

Overall, I find this to be a very interesting and thought-provoking manuscript. Basically, the authors have identified a variety of dinucleotides as 5' RNA caps in bacteria, and then they have identified a very interesting regulatory mechanism – the presence of single methyl groups as inhibitors of capping. This concept was previously introduced in mammalian mRNA, but the current paper shows a completely new and interesting way that this could occur. In particular, they show that the methyl modification inhibits RppH by preventing a cation-pi interaction.

To me, the only key experiment that is missing, is a clear demonstration of the identity of the methyl modifications that are capable of inhibiting RppH. For example, the author said that they think it is either 2'-O or N7 methyls on G. I feel like this needs to be known, because otherwise the mechanism is too unclear. I recognize that this is complicated to do by mass spectrometry, but my question would be whether the authors can use a QQQ MS, with simple standards in order to use fragmentation spectra to get more confidence about the potential location of the methyl modifications. In this way, the authors can selectively make sure the half-life of methyl-modified caps vs. non-modified caps. If they can't do this in cells, they can at least repeat their in vitro experiments where they added RppH and measure the decay rate of each species, rather than to generically measure methylated caps. Notice, the effect might be much more dramatic if they were able to selectively measure the 2'-O or the N7 methyl forms – only one of these might be resistant to RppH. Overall, I'm not expecting any specific experiment, but I am suggesting that the authors decide on a way to get more confidence about the specific methyl modifications, and then include data that can support the claim that that specific methyl modification makes the RNA uniquely resistant to RppH.

Other issues are as follows:

1. In some of the experiments, the authors use T7 RNA polymerase and determine whether or not the nucleoside phosphates can be used as the initiating nucleotide. Although the experiments with E. coli RNA polymerase are important, because they reflect the potential use of these dinucleotides can be used for endogenous transcription, the studies with T7 are a little less physiologically relevant. I think the manuscript needs to justify why T7 was tested. I'm not saying that this should be removed, but I think the authors need to do a better job in explaining what was their reasoning, and how the reader should interpret these results especially in light of the very limited use of this enzyme in actual physiology, rather than as a molecular biology tool.

2. There has recently been considerable concern that preparations of RNA can contain NAD and other dinucleotides, even without any RNase treatment (despite size exclusion experiments). Therefore side-by-side analysis is needed in all MS experiments with undigested RNA. I suspect some of the signal is due to dinucleotides that are not part of RNA. This could affect the conclusions about which dinucleotides are in RNA.

Minor

1. One interesting analogy to this work is the previous work by Mauer et al., Nature 2017, in which they showed that mammalian mRNA caps that contain m⁷G-ppp-m⁶Am are efficiently decapped by Dcp2. I believe this 2017 paper is the first paper that documents that single methyl groups at the cap can regulate decapping. I think the current manuscript is an interesting extension of this concept. I think this would certainly be interesting to compare the current findings with this previous study in 2 or 3 sentences.

ÚOCHB ^{AV}
^{ČR}
IOCB PRAGUE

Ústav organické chemie a biochemie
Akademie věd České republiky, v. v. i.
Institute of Organic Chemistry and Biochemistry
of the Czech Academy of Sciences

We would like to thank the reviewers for their positive criticism and for pointing out many issues that we addressed to improve the quality of the presentation of our results.

Based on the suggestions from the reviewers and based on new experiments, we restructured the manuscript and we highlighted only the key new results. All other changes are not highlighted for better orientation, but they are discussed in the detailed answers to reviewers.

Reviewers' comments:

Reviewer #1 (Remarks to the Author):

Using LC-MS, Hudecek et al. provide evidence for the presence of both methylated and non-methylated short NpnN-capped RNAs in E.coli cells. They further demonstrate that RppH (E.coli 5' pyrophosphohydrolase) efficiently cleaves dinucleoside polyphosphates from the RNA and its activity is inhibited by NpnN methylation. Interestingly, the authors found that amounts of NpnNs and their methylation increased in the stationary phase of bacteria growth (stress condition) and propose that bacteria use methylated caps to stabilize some RNAs under stress. The manuscript presents interesting findings, however the framework of this manuscript needs to be substantially improved.

Answer:

We are very encouraged by the positive criticism of the reviewer and made huge efforts to substantially improve the manuscript.

Specific comments:

1. The authors detected Ap3A, Ap3G and Ap5A as well as methyl-Ap3A, dimethyl-Gp4G and methyl-Ap5G caps in exponential phase and additionally methyl-Ap4G, methyl-Ap5A and dimethyl-Ap5G in the stationary phase by LC-MS methodology. It is crucial to identify where the methylation sites are within the cap structure. The entire manuscript is premised on the first nucleotide being the methylation sites, including the proposed resistance to RppH cleavage. The authors could use chemically synthesized standards and LC-MS/MS method to identify and quantify methyl-modified dinucleotide polyphosphates.

Answer:

To decipher the methylation positions, it is necessary to have available synthetic standards. The authenticity of the detected structure can be confirmed by the identical retention time or by fragmentation studies using QTRAP. The synthetic standards are used for defining the best fragmentation conditions that are later applied to the sample prepared from the isolated RNA. If the fragmentation pattern of the standard and the detected compound is the same, the chemical structure can be confirmed. We decided to identify methylation positions in 2mGp₄G and mAp₃A.

We used custom-synthesized m^7Gp_4Gm , m^1Gp_4Gm and m^2Gp_4Gm (Jena Bioscience), as we presumed that the structure of $2mGp_4G$ can be similar to eukaryotic m^7Gp_3Nm cap. And we synthesized (synthesis and characterization of compounds are part of the revised SI) all potential methylated Ap_3A : m^6Ap_3A , m^1Ap_3A and Amp_3A . Based on the retention time (Supplementary Fig. 10) we successfully identified m^7Gp_4Gm as one of the identified caps. As Amp_3A and m^6Ap_3A have a similar retention time and m^1Ap_3A undergoes Dimroth rearrangement under basic conditions (measurement conditions pH 9-10) to m^6Ap_3A , we analyzed samples using QTRAP. Based on the retention time and the fragmentation pattern, we were able to confirm that the samples contain m^6Ap_3A (Supplementary Fig. 11). Nevertheless, we cannot completely exclude the presence of small portion of m^1Ap_3A , as it can undergo Dimroth rearrangement to m^6Ap_3A . However, the option that there are two types of methylated Ap_3A caps seems rather improbable to us.

There are too many possibilities, where e.g. mAp_5G can be methylated, that the synthesis of all the standards would be too laborious and time demanding and definitely out of scope of this work. We decided to take a different approach – methyltransferase mutant analysis, which will be the matter of a future study.

2. How abundant are NpnN-capped (and methylated) RNAs? What are their levels compared to RNAs capped with other non-canonical initiating nucleotides, for instance NAD or just uncapped triphosphate RNA? What is the relative ratios of the different dinucleotide caps?

Answer:

The absolute quantification is possible only for the caps, where the synthetic standards are available. We quantified the amount of Ap_3A , Ap_3G , Ap_5A and m^7Gp_4Gm and we compared it with NAD and CoA. Ap_3A , Ap_3G and Ap_5A are present in similar concentrations as CoA approx. 75 fmol/ μg sRNA. The concentration of m^7Gp_4Gm is rather comparable to NAD. We observed 1200 fmol/ μg sRNA of m^7Gp_4Gm and close to 1900 fmol/ μg sRNA of NAD in the late stationary phase. (Fig. 2i and Supplementary Fig. 12).

3. Figure 4a should be removed. This is a simulation that may or may not have any relevance to the

actual structure. Extensive mutagenesis should be carried out to validate the conserved contacts with corresponding in vitro functional studies before this figure is presented in the manuscript.

Answer:

After long and deliberate considerations of the two comments of the reviewers (1, 2), we decided to leave (updated!) Fig. 4a in the manuscript. We fully agree that the role of molecular dynamics calculations is supportive and they do not provide direct evidence of the experimentally observed phenomena, due to the sheer complexity of the problem studied. They might provide, though, a graphical representation of that complexity and perhaps guide the reader throughout the manuscript by having the structural (graphical) information in mind. Namely, based on MD simulations we anticipated on the key role of the Arg residue in substrate binding and proposed the N7-methylation of the guanine, which introduces a positive charge to the purine ring system, to disrupt this interaction. The experimental results added to the manuscript in the revision phase confirm *a posteriori* the position of the methyl group and thus also predictive potential of the MD simulations.

4. Why did the authors focus only on short RNAs? Does it mean that no NpnNs were not detected in the long RNA pool?

Answer:

We also detected Np_nNs in long RNA. Nevertheless, the longer RNA, the lower the ratio of any cap to canonical nucleotides after the RNA digestion. I.e. the noise to signal ratio in the LC-MS measurement is higher and the detection of new *m/z* is thus hampered.

5. page 1, the authors made a point that T7 RNAP can efficiently incorporate NpnN during in vitro transcription. It has previously been reported that Gp4G, Gp5G and their methylated counterparts can be efficiently utilized by T7 polymerase (Jemielity at al.RNA 2003). This should be cited.

Answer: We added the citation on page 2.

“The only enzymatic incorporation of an Np_nN into RNA that had been reported previously involved methylated derivatives of Gp_nG (eukaryotic cap variants) prepared for a translation inhibition study (Jemielity at al.RNA 2003).”

6. Page 2, the extended data Fig 1c shows 33% capping for the ATP/NAD = 1 which is comparable to NpnN. If this statement is true, it does not support the sentence “the NAD-capped RNA was produced in lower amounts compared to the majority of NpnNs”.

Answer:

We changed the statement to: “The NAD-capped RNA was produced in a comparable amount to majority of Np_nNs.”

7. Page 5, Figure legend 3, it would be beneficial to include the length of RNA, how the RNA was labeled, and a % of page gel used in the experiment. Additionally, in panel d and e, it should be indicated that exonuclease was present in the reaction as well.

Answer:

We added the description in Fig. legend 1 and 3. Concerning kinetic studies with RppH and ApaH (in previous version panel 3d,e), any exonuclease (terminator) was not used in the experiment and we calculated the disappearance of the capped RNA.

8. The authors should check spelling and words used throughout the manuscript. For example: page

5 “we added RppH to transform the 5'-capped RNA into 5'-p RNA..” Not sure if this would be considered a transformation.
“The decapping reaction of Ap_{4,5}N RNA was almost quantitative.” what does almost quantitative mean?

Answer:

We changed the verb to transform for to cleave and we rephrased the expression “The decapping reaction of Ap_{4,5}N RNA was almost quantitative.” To “The decapping reaction of Ap_{4,5}N RNA was efficient and within 5 min around 80% of capped RNA was cleaved.”

Reviewer #2 (Remarks to the Author):

The manuscript by Hudeček et al. provides evidence for the presence of a previously unknown 5' end modifications on RNA isolated from E. coli cells. In particular, evidence is presented that small RNAs isolated from E. coli contain 5' end dinucleoside polyphosphates (some of which also contain one or two methyl groups). To detect these 5' end structures, the authors use LC-MS methods previously used by David Liu's group in work published in 2009 to detect 5'-NAD caps and 5'-CoA caps on RNAs isolated from E. coli. In addition to the LC-MS results, the authors provide two additional pieces of information. First, experiments that show both E. coli RNA polymerase and T7 RNA polymerase can use dinucleoside polyphosphates as substrates for transcription initiation in vitro (thus providing a mechanistic explanation for the presence of 5'-end dinucleoside polyphosphate structures on RNAs isolated from cells). Second, in vitro experiments that show E. coli RppH can process RNAs containing a 5'-end dinucleoside polyphosphates, leaving a 5' monophosphate.

In principle, the finding that 5'-end dinucleoside polyphosphate “caps” that resemble the structure of the eukaryotic m7G cap are present on RNAs isolated from E. coli is suitable for publication in a top journal. However, the current version of the manuscript needs substantial revisions before being appropriate for publication.

Answer:

We are very encouraged by the positive criticism of the reviewer and made huge efforts to substantially improve the manuscript.

Specific changes requested:

(1.) The thesis work should be cited and/or discussed: “Stress-induced capping of mRNA in bacteria” by Daniel Luciano from Joel Belasco's lab (NYU, 2014). The results presented in this thesis suggest particular full-length mRNAs that contain 5' dinucleoside polyphosphate “caps” and suggest the involvement of ApaH rather than RppH in the processing of these cap structures.

Answer:

Unfortunately, the thesis has not been publicly available. We do, however, cite the work of Belasco and coworkers, which was submitted in Mol. Cell concurrently with our work.

(2.) Claims regarding the relative efficiencies of dinucleoside polyphosphates vs. NTPs as initiating nucleotides should be removed from the manuscript unless the appropriate kinetic competition assays are performed. The claim that “NpnNs are superior initiating substrates compared to NAD,” should be removed from the manuscript unless the appropriate kinetic competition assays are

performed. Furthermore, NAD-capping (and presumably use of NpnNs as initiating nucleotides) exhibits strong promoter sequence specificity. Thus, any claim regarding efficiency of NpnNs vs. NAD must take into consideration the promoter templates used for the comparison. Analysis of relative efficiencies may also require use of A-less or G-less templates depending upon the NpnN analyzed.

Answer:

We agree. We varied the concentrations of the initiating non-canonical substrates against fixed concentrations of ATP or GTP and it is stated in the manuscript. The results revealed how well these non-canonical substrates were able to compete with the natural substrates, and this allowed to compare the non-canonical substrates also among themselves under the studied conditions (including the promoter sequence). Therefore, we modified the text:

"To identify the best substrate for RNAP under *in vitro* conditions, we varied..."

"The similar behavior of Np_nNs to NAD supports the theory that these molecules might be present as new 5'-RNA caps in cells."

Furthermore, we removed the sentence: "Np_nNs are superior initiating substrates compared to NAD," and changed it for the sentence: "We used two different model, well characterized promoters (*rrnB* P1 and *rnaI*) for *E. coli* RNA polymerase. In both cases, we observed a higher production of Np_nN-RNA compared to NAD-RNA (Fig. 1d)."

As it is known for the promoter-dependence of the NAD, NADH and CoA incorporation into RNA – it is definitely possible that it plays a role also for incorporation of Np_nNs, and we are currently performing extensive experiments addressing this issue. The result will be reported in due time. In the manuscript, we now mention that the promoter sequence may affect the outcome ("Nevertheless, the promoter sequence may affect the outcome as it is known for NAD.").

Nevertheless, we included a new experiment using the *rnaI* promoter in *in vitro* transcription with *E. coli* RNA polymerase. We added the primary data into panel 1d. The results (relative efficiencies of non-canonical substrate initiation) were similar to those obtained with the *rrnB* P1 promoter (Fig. 1d): under the conditions used in the experiment we detected relatively low levels of NAD-RNA.

(3.) The authors analyze removal of dinucleoside polyphosphate caps by RppH. Authors should comment on why they did not test NudC (previously shown to remove various adenosine-containing RNA-caps such as NAD), ApaH (which is proposed to remove dinucleoside polyphosphate caps in the work reported in the Luciano lab thesis), or other Nudix-family members in *E. coli*. Without analysis of these other enzymes, it is unclear whether RppH's activity *in vitro* reflects a role in the removal of dinucleoside polyphosphate caps *in vivo* (especially given no differences in dinucleoside polyphosphate caps are observed by LC-MS analysis when RppH is removed from cells). In addition, both unmethylated and methylated caps should be analyzed for sensitivity to NudC and ApaH.

Answer:

We purified NudC and ApaH and tested both enzymes on Np_nN-RNAs. The results are added to manuscript in Fig. 3 and Supplementary Fig. 13-16.

(4.) Many changes in Figures and legends should be made. Where appropriate, figure legends should indicate the number of technical replicates and/or number of biological replicates, and indicate any

statistical analysis used. Figure 1e is very confusing and the graph format and color choices make the data essentially uninterpretable. The table in figure 2b should be split into separate tables to make the data that is presented clear. For figures 3b and c, state which RNA polymerase, template, and NTPs were used to generate the RNA products.

Answer:

We added the requested information into the legends. We changed colours in Fig. 1e and we increased the size of the panel. We split the tables in Fig. 2b.

(5.) The authors should justify the choice of the *rrnB* P1 promoter for the in vitro transcription experiments using *E. coli* RNAP. *rrnBP1* is an atypical promoter in terms of its biochemical properties (unusual start site, unstable open complexes). Ideally, the authors would provide experiments analyzing capping on at least one standard promoter (such as *lacUV5*). The authors should also provide sequence information for the promoter templates analyzed in vitro in the appropriate figures. Also, it was unclear how experiments using T7 RNAP fit into this story given that T7 RNAP is not present in *E. coli* cells from which these caps are detected.

Answer:

We are thankful for the suggestion.

rrnB P1 is a model, well characterized promoter. It is strong promoter when it is supplied with sufficient amounts of the initiating substrate. This condition is easily satisfied *in vitro*, hence we used it for these experiments. Nevertheless, prompted by the reviewer, we tested also another promoter. As *lacUV5* is a synthetic promoter, we chose the *rnaI* promoter instead. This promoter is well-characterized and frequently used in our laboratories. Moreover, the *rnaI* promoter is a naturally occurring promoter. Finally, the RNAI was reported to contain the NAD cap *in vivo*. Therefore, we added an experiment with the *rnaI* promoter (Fig. 1d). The promoter sequence information is now part of the Fig. 1.

We used T7 RNAP as our model enzyme to test whether Np_nNs can be used as transcription initiating substrates. T7 RNAP is from the T7 phage, which infects most *E. coli* strains, and encounters the same intracellular milieu as the host enzyme. The use of T7 RNAP and *E. coli* RNAP also shows that the capability of RNAPs to accept Np_nNs as non-canonical initiating nucleotides is more general rather than being specific for one particular RNA polymerase. T7 RNAP was then used to provide the first comparative glimpse of Np_nN utilization due to the strong, reliable signal it provided.

(6.) For Figure 3c, The RNA products in the second and third lanes from the left of the gel do not show the expected migration as observed in panel b. Have the authors mislabeled the gel? The authors should address why in Figure 3c it appears the RNA product generated in the lane containing NAD^+ does not appear to carry an NAD-cap capped RNA. Samples in Figure 3 were DNase I treated using a protocol that involves heat inactivation. NAD is a heat-labile, therefore results obtained with NAD may have been affected by inclusion of a high temperature step.

Answer:

We are thankful for the comment. Concerning Fig. 3c, we mislabeled the gel. It is corrected in revised version (now, moved to Supplemental data).

Concerning NAD, the RppH does not cleave the NAD-RNA under given conditions. Therefore, after the terminator treatment one band disappears (corresponding to 5'-p RNA) and one stays intact

(corresponding to NAD-RNA). Otherwise, the NAD-RNA and 5'-ppp RNA migrate similarly on 12% PAGE. Although, the NAD is susceptible to thermal degradation, we did not observe any degradation. We heat the RNA at 75°C for 10 min. Partial degradation was reported for 85°C (Hachisuka, Journal of Bacteriology, 2017).

(7.) For Figure 4c, the authors should better describe and provide a sufficient justification for the inclusion of this molecular dynamics modeling. It was not clear precisely how the authors assigned the location of the methyl groups within NpnN caps. Also, the structure shown in the figure panel is uninformative as the color scheme used makes it difficult to see features of the structure (Mg ions are the same color as amino acids and Gp4G; the orange and red colors are very similar as well).

Answer:

As mentioned in the answer to the Reviewer 1, we decided to include molecular dynamics simulations because they provide molecular basis for the enriched presence of methylated Np_nN caps *in vivo* and graphical and structural guidance for the reader. Based on MD simulation with Gp₄G, we realized that the π -stacking of guanine ring system with positively charged arginines is crucial for the substrate binding and proposed the N⁷ methylation of the guanine, because the positive charge would disrupt these interactions (as can be seen in the corresponding simulation). In the revised version of the manuscript, we were able to confirm the position of the methyl groups experimentally. We believe that we have thus demonstrated the predictive power of molecular modelling which provides a justification for presenting the MD results as a qualitative but illustrative bit to the overall picture. Obviously, we have not described this full *genesis* into the revised version of the manuscript, but we thank the reviewers for their comments concerning the need for MD simulations that lead us to thorough experimental investigations - justifying them (MD) *a posteriori*. We believe they now represent an integral part of the revised manuscript. To improve the clarity of Fig. 4b, we have changed the color scheme as suggested by the reviewer.

(8) The authors mention a heat shock growth condition in the method section that is not mentioned elsewhere in the main body of the paper.

Answer:

We are thankful for the comment. We removed that part.

(9) Authors should comment on whether any of the signals derived from 5' end modifications of unknown origin presented in the supplemental data of the prior 2009 work from Liu and co-workers correspond to the 5' ends detected in this work.

Answer:

We compared our detected *m/z* with those reported by Liu et al. in 2009. We found that the *m/z* 771.0726 could correspond to Ap₃G detected by us. Otherwise, any of detected Np_nNs was not reported by Liu et al.

We added these sentences into the text:

"We compared our detected *m/z* signals with those reported by Liu (Chen, Nat Chem Biol 2009). The only similar *m/z* was 771.0726, which can correspond to Ap₃G."

(10) The authors should comment on why the LC-MS analysis presented in the paper was performed using RNA from DH5alpha cells, but the rppH knockout strain analyzed was generated in a different strain. Why not analyze DH5alpha cells that contain the rppH deletion?

Answer:

We decided to remove this part from our article as it does not bring any new information.

To clarify the cell choice: We had rppH knockout strain available and when we compared the DH5 α and wild type, we did not observe any difference in the detected Np_nNs. As the RNA isolated from the mutants and wild type did not show any differences in Np_nN caps levels, we decided not to continue in experiments and not to prepare new mutants.

(11) For Figure 3, panels d and e, it may be difficult to analyze the effect of RppH on ppp-RNA due to the difficulty in separating ppp-RNA and p-RNA using the electrophoresis conditions used.

Answer:

The separation of the ppp-RNA and p-RNA was very good under the used conditions and the quantification was not influenced. We add the gel illustrating that.

Reviewer #3 (Remarks to the Author):

Overall, I find this to be a very interesting and thought-provoking manuscript. Basically, the authors have identified a variety of dinucleotides as 5' RNA caps in bacteria, and then they have identified a very interesting regulatory mechanism – the presence of single methyl groups as inhibitors of capping. This concept was previously introduced in mammalian mRNA, but the current paper shows a completely new and interesting way that this could occur. In particular, they show that the methyl modification inhibits RppH by preventing a cation- π interaction.

To me, the only key experiment that is missing, is a clear demonstration of the identity of the methyl modifications that are capable of inhibiting RppH. For example, the author said that they think it is either 2'-O or N7 methyls on G. I feel like this needs to be known, because otherwise the mechanism is too unclear. I recognize that this is complicated to do by mass spectrometry, but my question would be whether the authors can use a QQQ MS, with simple standards in order to use fragmentation spectra to get more confidence about the potential location of the methyl modifications. In this way, the authors can selectively make sure the half-life of methyl-modified caps vs. non-modified caps. If they can't do this in cells, they can at least repeat their in vitro experiments where they added RppH and measure the decay rate of each species, rather than to generically measure methylated caps. Notice, the effect might be much more dramatic if they were able to selectively measure the 2'-O or the N7 methyl forms – only one of these might be resistant to RppH. Overall, I'm not expecting any specific experiment, but I am suggesting that the authors decide on a way to get

more confidence about the specific methyl modifications, and then include data that can support the claim that that specific methyl modification makes the RNA uniquely resistant to RppH.

Answer:

As mentioned in our answer to Reviewer #1, we used custom-synthesized standards of 2mGp₄G: m⁷Gp₄Gm, m¹Gp₄Gm and m²Gp₄Gm (Jena Bioscience). Based on the retention time, we were able to identify the structure of the cap as m⁷Gp₄Gm. We attempted to incorporate this cap into RNA by *in vitro* transcription. Nevertheless, all dimethylated Gp₄Gs were very poor substrate for T7 RNAP (Supplemental Fig. 17). Therefore, the *in vitro* studies of model RNA with decapping enzymes were not possible. MD shows that the N⁷ methylation causes a loss of interactions with the RppH active site and thus hampers the cleavage. The detailed study of the methylation effects will be possible, once we know the methyltransferases responsible for the Np_nNs-RNA methylations. This will be matter of our future study. Nevertheless, we discovered another decapping enzyme – ApaH that is capable to cleave even m⁷Gp₄Gm-RNA (Fig. 4a).

Other issues are as follows:

1. In some of the experiments, the authors use T7 RNA polymerase and determine whether or not the nucleoside phosphates can be used as the initiating nucleotide. Although the experiments with *E. coli* RNA polymerase are important, because they reflect the potential use of these dinucleotides can be used for endogenous transcription, the studies with T7 are a little less physiologically relevant. I think the manuscript needs to justify why T7 was tested. I'm not saying that this should be removed, but I think the authors need to do a better job in explaining what was their reasoning, and how the reader should interpret these results especially in light of the very limited use of this enzyme in actual physiology, rather than as a molecular biology tool.

Answer (same as Answer 5 to Reviewer #2)

We wanted to show that the capability of RNA polymerases to accept Np_nNs as non-canonical initiating nucleotides is rather general and not specific for one particular RNA polymerase. Moreover, the high performance of T7 RNAP helped us to compare better the Np_nN between each other.

2. There has recently been considerable concern that preparations of RNA can contain NAD and other dinucleotides, even without any RNase treatment (despite size exclusion experiments). Therefore side-by-side analysis is needed in all MS experiments with undigested RNA. I suspect some of the signal is due to dinucleotides that are not part of RNA. This could affect the conclusions about which dinucleotides are in RNA.

Answer:

We performed negative controls (untreated with Nuclease P1) for every experiment. The negative controls are now added to SI (Supplemental Fig. 2-4).

Minor

1. One interesting analogy to this work is the previous work by Mauer et al., Nature 2017, in which they showed that mammalian mRNA caps that contain m⁷Gppp-m⁶Am are efficiently decapped by Dcp2. I believe this 2017 paper is the first paper that documents that single methyl groups at the cap can regulate decapping. I think the current manuscript is an interesting extension of this concept. I

think this would certainly be interesting to compare the current findings with this previous study in 2 or 3 sentences.

Answer:

We added this comment into the Discussion section: “In human cells, it has been shown that the N^6 -methylation of the first encoded nucleotide (m^6Am , m^6A) hampers the cleavage of mRNA by Nudix decapping enzyme Dcp2 (Mauer et al. Nature 2016). Our proposed mechanism correlates with this finding and suggests that the strategy, by which cells protect their RNA against decapping by methylation, is general and may also be found in higher organisms.”

REVIEWERS' COMMENTS:

Reviewer #1 (Remarks to the Author):

The authors have addressed my initial concerns.

Reviewer #2 (Remarks to the Author):

The novelty of the work reported in this manuscript has been diminished by recent findings reported by the Belasco lab (Mol Cell, Sept 2019). However, the results presented by Hudecek et al. do still provide information that will be of interest to the field. In particular, the LC-MS analysis by Hudecek et al. provide evidence for non-methylated and methylated Np_nN caps on RNAs isolated from cells that, unlike in the Belasco study, have not been subjected to Cadmium stress. Critically, the structures of the RNA caps reported in this work (detected in RNA isolated from cells not subjected to Cadmium stress) differ from the structures of the RNA caps reported in the Belasco work (detected in RNA isolated from cells subjected to Cadmium stress). Thus, the results of the LC-MS analysis reported in this manuscript build upon the results reported in the Belasco study and are of use to the field. In addition, the results in this work showing RppH processes non-methylated caps but not methylated caps are also of interest.

Although it is not clear what functional role the RNA caps reported in this work play in cells **the work is appropriate to publish provided the two modifications to the text listed below are made.**

First, the last two sentences of the abstract overstate the significance of the work and also misrepresent the results provided in this paper. As mentioned above, the novelty of the findings reported here have been diminished by published work from the Belasco group. The phrases in *italics* are inaccurate given the work from Belasco while the phrase in **bold** is speculative. These sentences should be rewritten to accurately portray how these results relate to what has already been reported and what evidence is provided in this paper.

Our work introduces *an entirely new perspective* on the chemical structure of RNA in prokaryotes and on the role of RNA caps. It is also *the first evidence* that small molecules such as Np_n **are incorporated into RNA and thus influence the cellular metabolism and RNA turnover**.

Second, the authors should add a section that clearly explains how the findings reported in this work are similar to or differ from those reported in the Belasco work. The authors should provide potential explanations to account for the differences (for example, differences in cell growth, differences in methods of detection of caps, etc...).

Reviewer #3 (Remarks to the Author):

Overall, I think this manuscript is improved with the new changes.

However, I suggest one important textual change/clarification: The authors start to explore whether T7 polymerase can use non-canonical initiating nucleotides. However, T7 polymerase has little relevance to endogenous polymerases made by bacteria. The authors should explain why T7 and its properties would be relevant or teach us about the properties of bacterial polymerases. (The authors talk about T7's low error rate and high efficiency - but this is not relevant to whether bacterial polymerases would show use non-canonical initiators such as the ones tested. If the behavior of T7 polymerase is not related to other polymerases, then the T7 experiments would be a useless and non-physiologically relevant experiment. As an example, they should talk about the initiation mechanisms and the nucleotide-binding pocket for initiation for T7 and bacterial RNA polymerases. They should do this before they start using or discussing data with T7.

ÚOCHB AV
IOCB PRAGUE

Ústav organické chemie a biochemie
Akademie věd České republiky, v. v. i.
Institute of Organic Chemistry and Biochemistry
of the Czech Academy of Sciences

We would like to thank the reviewers for their suggestions and contributions to the peer review of our manuscript.

REVIEWERS' COMMENTS:

Reviewer #1 (Remarks to the Author):

The authors have addressed my initial concerns.

Reviewer #2 (Remarks to the Author):

The novelty of the work reported in this manuscript has been diminished by recent findings reported by the Belasco lab (Mol Cell, Sept 2019). However, the results presented by Hudecek et al. do still provide information that will be of interest to the field. In particular, the LC-MS analysis by Hudecek et al. provide evidence for non-methylated and methylated Np_nN caps on RNAs isolated from cells that, unlike in the Belasco study, have not been subjected to Cadmium stress. Critically, the structures of the RNA caps reported in this work (detected in RNA isolated from cells not subjected to Cadmium stress) differ from the structures of the RNA caps reported in the Belasco work (detected in RNA isolated from cells subjected to Cadmium stress). Thus, the results of the LC-MS analysis reported in this manuscript build upon the results reported in the Belasco study and are of use to the field. In addition, the results in this work showing RppH processes non-methylated caps but not methylated caps are also of interest.

Although it is not clear what functional role the RNA caps reported in this work play in cells **the work is appropriate to publish provided the two modifications to the text listed below are made.**

First, the last two sentences of the abstract overstate the significance of the work and also misrepresent the results provided in this paper. As mentioned above, the novelty of the findings reported here have been diminished by published work from the Belasco group. The phrases in italics are inaccurate given the work from Belasco while the phrase in bold is speculative. These sentences should be rewritten to accurately portray how these results relate to what has already been reported and what evidence is provided in this paper.

Our work introduces *an entirely new perspective* on the chemical structure of RNA in prokaryotes and on the role of RNA caps. It is also *the first evidence* that small molecules such as Np_nN are **incorporated into RNA and thus influence the cellular metabolism and RNA turnover.**

Second, the authors should add a section that clearly explains how the findings reported in this work

are similar to or differ from those reported in the Belasco work. The authors should provide potential explanations to account for the differences (for example, differences in cell growth, differences in methods of detection of caps, etc...).

Answer:

We do not agree with the statement of reviewer #2 that our work was diminished by the work of Belasco (Mol Cell, Sept 2019). Our work was published on the preprint server bioRxiv (doi.org/10.1101/563817) on February 28, 2019, while Belasco's work was submitted for publication in Mol. Cell on March 20, 2019.

We modified the last two sentences in the abstract:

“Our work introduces a different perspective on the chemical structure of RNA in prokaryotes and on the role of RNA caps. We bring evidence that small molecules such as Np_nNs are incorporated into RNA and may thus influence the cellular metabolism and RNA turnover.”

We added a comment for the discrepancy of detected caps by us and by Belasco:

“However, we did not detect any of the caps reported therein by our LC-MS technique, as the Ap₄N-RNA caps were detected under different stress conditions using a different detection technique.”

Reviewer #3 (Remarks to the Author):

Overall, I think this manuscript is improved with the new changes.

However, I suggest one important textual change/clarification: The authors start to explore whether T7 polymerase can use non-canonical initiating nucleotides. However, T7 polymerase has little relevance to endogenous polymerases made by bacteria. The authors should explain why T7 and its properties would be relevant or teach us about the properties of bacterial polymerases. (The authors talk about T7's low error rate and high efficiency - but this is not relevant to whether bacterial polymerases would show use non-canonical initiators such as the ones tested. If the behavior of T7 polymerase is not related to other polymerases, then the T7 experiments would be a useless and non-physiologically relevant experiment. As an example, they should talk about the initiation mechanisms and the nucleotide-binding pocket for initiation for T7 and bacterial RNA polymerases. They should do this before they start using or discussing data with T7.

Answer:

We thank the Reviewer #3 for his/her advice and suggestion. We removed our previous statements explaining the use of T7 RNAP. Now, we explain the T7 RNAP study relevance in the following sentences:

“T7 RNAP was selected as a first tool to explore the capability of Np_nNs to be substrates of RNAPs as it was previously shown to be able to use comparably sized non-canonical initiating substrates. Consistently, based on 3D structures, the nucleotide-binding pockets for initiation phase of T7 and *E. coli* RNAP are spacious enough to accommodate such substrates.”

Yours sincerely,

Hana Cahová, Ph.D. (on behalf of all authors)